# TRANSFORMERS LEARN MOLECULAR STRUCTURES WITHOUT GRAPH PRIORS

## ABSTRACT

Graph Neural Networks (GNNs) are the dominant architecture for molecular machine learning, particularly for molecular property prediction and machine learning interatomic potentials (MLIPs). GNNs perform message passing on predefined graphs often induced by a fixed radius cutoff or $k$-nearest neighbor scheme. While this design aligns with the locality present in many molecular tasks, a hard-coded graph can limit expressivity due to the fixed receptive field and slows down inference with sparse graph operations. In this work, we investigate whether pure, unmodified Transformers trained directly on Cartesian coordinates—without predefined graphs or physical priors—can approximate molecular energies and forces. As a starting point for our analysis, we demonstrate how to train a Transformer to competitive energy and force mean absolute errors under a matched training compute budget, relative to a state-of-the-art equivariant GNN on the OMol25 dataset. We discover that the Transformer learns physically consistent patterns—such as attention weights that decay inversely with interatomic distance—and flexibly adapts them across different molecular environments due to the absence of hard-coded biases. The use of a standard Transformer also unlocks predictable improvements with respect to scaling training resources, consistent with empirical scaling laws observed in other domains. Our results demonstrate that many favorable properties of GNNs can emerge adaptively in Transformers, challenging the necessity of hard-coded graph inductive biases and pointing toward standardized, scalable architectures for molecular modeling.

## 1 INTRODUCTION

Graph Neural Networks (GNNs) have been the dominant architecture for molecular property prediction. Especially for 3D geometric tasks, GNNs rely on a predefined graph construction algorithm for message passing along with strong physical inductive biases (Batzner et al., 2022; Batatia et al., 2022; Gasteiger et al., 2021). These inductive biases include custom featurization, such as geometric descriptors (Gasteiger et al., 2021), and explicitly built-in symmetries, like rotational equivariance (Batatia et al., 2022; Batzner et al., 2022; Fu et al., 2025; Liao et al., 2024). While some recent models challenge the necessity of built-in equivariance (Mazitov et al., 2025; Qu & Krishnapriyan, 2024; Neumann et al., 2024), they still add physics-inspired components to their model and still rely on a GNN as the backbone architecture. Broader molecular property prediction tasks have explored more varied architectures (Kim et al., 2022; Zhou et al., 2023; Eissler et al., 2025; Hussain et al., 2024), but still leverage physical inductive biases, such as graph-based embeddings.

The reliance on GNNs presents challenges when scaling to the vast chemical spaces and large computational-chemistry datasets available today (Chanussot et al., 2021; Barroso-Luque et al., 2024; Eastman et al., 2024; Schreiner et al., 2022; Levine et al., 2025). Theoretical limitations, including oversmoothing and oversquashing (Wu et al., 2023; Rusch et al., 2023; Giovanni et al., 2023; Topping et al., 2022), limit the expressivity of GNNs as their depth increases. The use of graphs can also lead to generalization problems (Bechler-Speicher et al., 2024), which was empirically found to also be an issue for molecular GNNs (Kreiman & Krishnapriyan, 2025). Practically, the sparse operations in GNNs complicate efficient training on modern hardware. Together, these drawbacks have made it challenging to train large graph-based models (Sriram et al., 2022) (see Section 2 for more details).

In contrast, the Transformer is the standard architecture in numerous other fields of machine learning (ML) (Dosovitskiy et al., 2021; Vaswani et al., 2023; Team et al., 2024; Brown et al., 2020; Touvron

et al., 2023a). The success of the Transformer has been guided by empirical scaling laws that precisely predict test performance based on dataset size, computational budget, and number of model parameters (Kaplan et al., 2020; Hoffmann et al., 2022). This common architectural framework, paired with established scaling laws, dramatically accelerates the research process by providing a standardized recipe for addressing a broad class of ML problems. As a result, it has enabled the development of powerful multi-modal models (et. al., 2024; Team et al., 2024), specialized hardware and software for Transformers (Ansel et al., 2024; Jouppi et al., 2017; Kwon et al., 2023), and beyond.

In this work, we use the large Open Molecules 2025 (OMol25) dataset as a case study (Levine et al., 2025) to investigate whether explicit physical inductive biases, including the graph itself, are necessary for accurately approximating molecular energies and forces. Drawing inspiration from the convergence to the Transformer architecture (Vaswani et al., 2023) in other fields of ML, we evaluate whether an off-the-shelf, unmodified Transformer can learn molecular energies and forces directly from Cartesian coordinates without relying on any physical inductive biases. Our findings reveal that, within the same training computational budget, Transformers can achieve the energy and force errors comparable to those of a state-of-the-art equivariant GNN on the new OMol25 dataset (Levine et al., 2025), while being faster at inference and training in wall-clock time. The use of the standard Transformer also enables scaling to 1B parameters with existing software and hardware, revealing consistent scaling laws that predict performance at scale. We explore the learned attention maps and discover that Transformers capture an inverse relationship between distance and attention strength. Transformers also learn to *adapt their effective receptive field*, attending to atoms farther away in less dense regions and concentrating attention to local interactions for tightly packed atoms.

While it remains crucial to assess the failure modes of MLIPs and their adherence to key physical principles before broad application to scientific discovery (Chiang et al., 2025; Deng et al., 2024; Deng, 2023; Kreiman & Krishnapriyan, 2025; Fu et al., 2023), our experiments suggest that, given the large chemical datasets now available (Levine et al., 2025; Chanussot et al., 2021; Barroso-Luque et al., 2024; Eastman et al., 2023; Schreiner et al., 2022; Deng, 2023), explicit graph-based inductive biases could potentially be learned directly from data. These results pave the way for transferring insights from the broader ML literature into the MLIP and molecular modeling community, leveraging a general and flexible architecture capable of addressing a wide range of chemical problems.

## 2 RELATED WORK

**Machine Learning Interatomic Potentials (MLIPs) and Molecular Property Prediction.** Machine learning interatomic potentials (MLIPs) are a popular application area for graph neural networks (GNNs). MLIPs are typically trained using supervised learning to predict a molecule-level energy and per-atom force labels, which are generated from reference computational chemistry methods (like Density Functional Theory). Behler & Parrinello (2007) popularized the use of MLIPs as a substitute for expensive computational chemistry calculations, leading to numerous applications of MLIPs for the study of chemical systems (Batatia et al., 2024; Garrison et al., 2023; Artrith & Urban, 2016). Many GNN-based MLIPs incorporate physical inductive biases into the architecture (Batatia et al., 2022; Gasteiger et al., 2021; Batzner et al., 2022), such as rotational equivariance and geometric features. Although recent models have started to discard some of these hard constraints (Qu & Krishnapriyan, 2024; Neumann et al., 2024; Mazitov et al., 2025), they continue to rely on GNNs as the architectural backbone. Eissler et al. (2025) also explore relaxing inductive biases in model design, but use a cubic attention mechanism that relies on the explicit calculation of pairwise displacements.

While some GNNs and MLIPs do incorporate attention-based mechanisms (Wu et al., 2023; Liao et al., 2024; Qu & Krishnapriyan, 2024), it is important to note *they are not using the standard Transformer architecture* (Vaswani et al., 2023) and still operate on a predefined graph structure. While a Transformer can be viewed as operating on a fully connected graph, its attention mechanism is fully connected for every input. In contrast, GNNs typically construct a new graph for each input (e.g., using a radius cutoff). Although some models for other molecular property prediction tasks have begun exploring Transformers, they either use only textual molecular descriptors (e.g., SMILES) (Chithrananda et al., 2020) or depend on custom graph embeddings (Kim et al., 2022; Rampášek et al., 2023) and modifications of the attention mechanism to include physical inductive biases (Zhou et al., 2023; Eissler et al., 2025). In this study, we instead explore using an unmodified Transformer without any graph structure or physics-inspired featurization.

**Scaling Laws.** Other fields of ML have converged on the Transformer architecture (Vaswani et al., 2023), including natural language processing (Devlin et al., 2019; et. al., 2024; Touvron et al., 2023a), computer vision (Dosovitskiy et al., 2021) and even robotics, where exact physical constraints are known (Team et al., 2024; Kim et al., 2024). A key factor driving this convergence is the discovery of empirical scaling laws, which reveal predictable relationships between validation loss, model size, dataset size, and computational resources (Kaplan et al., 2020; Henighan et al., 2020; Hoffmann et al., 2022). These laws have been observed over orders of magnitude in resources (Touvron et al., 2023a; Brown et al., 2020), showing that larger models reliably yield better performance given sufficient data and compute. In the context of MLIPs, Frey et al. (2023) explored scaling in neural network force fields but found significant deviations from consistent power law relationships with the models and datasets available at the time. More recently, Wood et al. (2025) found scaling trends on the new OMol25 dataset (Levine et al., 2025), but required a sophisticated mixture-of-experts scheme to train their largest models. To the best of our knowledge, there are no graph-based MLIPs at the scale of models seen in other ML fields in terms of number of parameters.

**Challenges With Graph-Based Learning.** Empirical evidence suggests GNNs are hard to scale compared to Transformers. Sriram et al. (2022) scaled a GemNet model up to 1B parameters but found the best performance was with only about 300M parameters. The MACE architecture also exhibits performance saturation at just two layers deep (Batatia et al., 2022; 2024; Kovács et al., 2023). Even recent models designed for scalability only have up to a few hundred million parameters when reporting their best results (Neumann et al., 2024; Qu & Krishnapriyan, 2024), which is still small in magnitude compared to current models in other fields of ML (Touvron et al., 2023a; Brown et al., 2020).

GNNs have a number of theoretical and practical issues that hinder their scalability. The permutation invariance and graph bottlenecks in message passing schemes can lead to oversmoothing (Rusch et al., 2023) and over-squashing (Giovanni et al., 2023; Topping et al., 2022) which theoretically limit the expressive power of GNNs at depth and hinder modeling long-range interactions (Dwivedi et al., 2023). These theoretical limitations also apply to graph-attention mechanisms (Wu et al., 2023). Bechler-Speicher et al. (2024) showed that the use of GNNs can hurt generalization when the targets are labeled without a graph structure. Kreiman & Krishnapriyan (2025) found that GNN-based MLIPs tend to overfit to the graph-structures encountered during training and struggle to generalize to new molecular geometries. The reliance on sparse operations across (potentially large) graphs further complicates efficient parallelization of training on modern hardware (Sriram et al., 2022; Gonzalez et al., 2012).

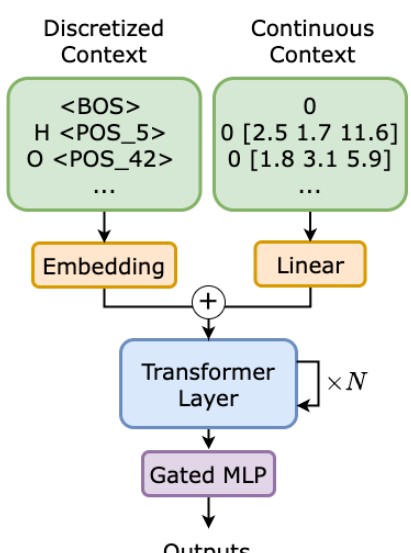

Figure 1: **Graph-Free Transformers Model Design.** Our model encodes both discretized and continuous molecular sequences using unmodified Transformer layers. Placeholder values represent discrete inputs in the continuous sequence.

## 3 TRAINING AN UNMODIFIED TRANSFORMER WITHOUT GRAPH PRIORS

To investigate the role of graph priors, we take machine learning interatomic potentials (MLIPs) as a representative case study (Unke et al., 2021). MLIPs learn to map three-dimensional atomic structures to molecule-level energies and per-atom forces, providing efficient surrogates for costly quantum-mechanical methods such as density functional theory (DFT). Traditionally, MLIPs incorporate physics-inspired features through graph-based message passing, where the molecular graph is constructed *a priori* using heuristics such as a fixed radius cutoff or $k$-nearest neighbors. While this approach aligns with the locality of many molecular interactions, it imposes a fixed receptive field and introduces computational overhead from sparse-graph operations.

In this work, we remove these constraints entirely by replacing the GNN with a **standard Transformer** operating directly on Cartesian coordinates—without predefined graphs or chemistry-specific

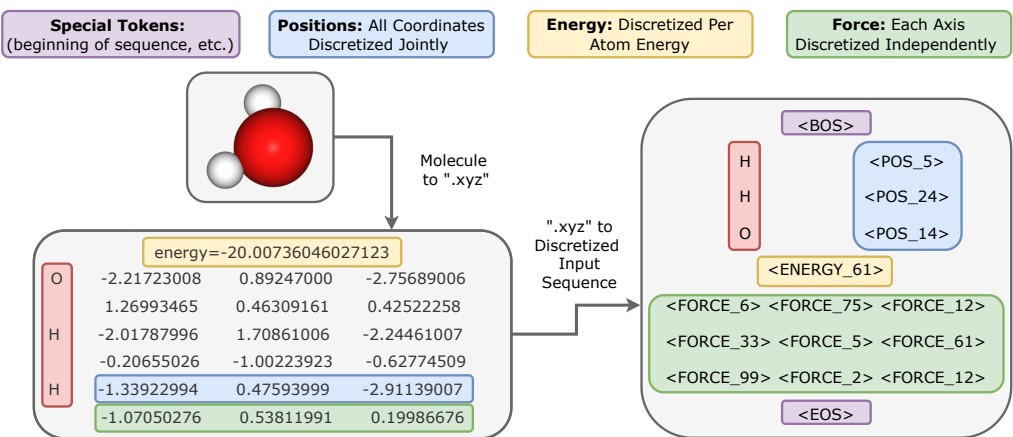

Figure 2: **Discretization Scheme during Pre-training.** We transform the standard ".xyz" molecular representation into a discretized input sequence for our model. To discretize continuous values, we use quantile binning, ensuring each bin contains the same number of datapoints. Atomic positions are jointly discretized into a 3D grid, while force and energy components are discretized independently along each dimension. We add special tokens, like beginning and end of sequence tokens. Note that these discretized tokens are accompanied by the continuous values (for positions, energies, forces, etc.), allowing the model to circumvent discretization errors for real-valued inputs.

architectural modifications. This provides a clean test bed for studying whether competitive molecular representations, and GNN-like relational patterns, can emerge naturally from data.

**Model Architecture.** We use the LLaMA2 architecture (Touvron et al., 2023b) as our backbone architecture, preserving the original multi-head self-attention mechanism. Our only architectural modifications are (1) removal of positional embeddings, since atomic positions are explicitly provided as input features and (2) the inclusion of both discrete and continuous embeddings for each token as model inputs, which are added before entering the attention layers. The attention mechanism itself remains *completely unchanged*, ensuring that any learned relational structure arises from the data rather than from built-in inductive biases. Figure 1 shows a schematic of the model architecture.

**Input Data Format.** Our model processes both a discretized and a continuous representation of each molecule's `.xyz` file. Continuous molecular features such as positions, forces, and energies often exhibit heavy-tailed distributions spanning multiple orders of magnitude (Figure 14), making using only a binned input prone to large discretization errors at the tails. Consequently, we give the model the continuous values on top of the discretized string, allowing it to rely on the continuous inputs for real-valued quantities.

As shown in Figure 2, we discretize continuous features using quantile binning so that each bin contains the same number of datapoints. The three position coordinates of each atom are jointly discretized into a three-dimensional grid, while force components are discretized independently along each axis. The continuous sequence provides the model with exact values for positions, forces, and energies, while using placeholder values to represent discrete inputs such as atomic numbers and special tokens. Special tokens mark the start and end of the sequence and indicate when the model should predict positions, forces, or energies. The embeddings for the continuous and discretized sequence are added before being input into the attention mechanism (see Appendix B for more details). An example of a discretized input string is shown in Figure 15.

**Training Procedure.** We adopt a two-stage training strategy inspired by pre-train–then–fine-tune approaches used by large language models (Touvron et al., 2023b; Brown et al., 2020; Hoffmann et al., 2022). During the pre-training stage, we train the model autoregressively with a causal attention mask to predict all discrete tokens in the sequence using a cross-entropy loss. This allows it to learn the joint distribution of positions, forces, and energies, enabling likelihood estimation (Appendix A), and providing a strong initialization for downstream fine-tuning.

During fine-tuning, the objective shifts to direct prediction of continuous energies and forces. We replace the causal attention mask with a bi-directional one, making the model permutation equivariant. We also replace the linear readout head that outputs a distribution over discrete tokens with two energy and force readout heads. The force head directly regresses the atomic embeddings to predict a force vector in $\mathbb{R}^3$ for each atom. The energy head predicts a per-atom energy which is aggregated across the system to get the total energy. During fine-tuning, the model operates in a continuous space, and no discretized tokens for energies or forces are included in the input.

# 4 WHAT CAN GRAPH-FREE TRANSFORMERS LEARN?

We base our investigation on the recently released OMol25 dataset (Levine et al., 2025), which provides energy and force labels calculated at the $\omega$B97M-V/def2-TZVPD level of theory. OMol25 covers a broad range of chemical structures, from biomolecules to metal complexes to electrolytes. The diversity and abundance of data makes it an ideal place to study molecular representation learning *without* graph priors.

Our analysis proceeds in three parts. First, we compare a large Transformer model to a state-of-the-art equivariant graph neural network (GNN) on energy and force prediction (Section 4.1). We then examine how Transformer performance scales with data and compute (Section 4.2). Finally, we analyze the learned representations of our graph-free model to understand what structural and physical information it captures (Section 4.3).

## 4.1 COMPARISON TO AN EQUIVARIANT GNN ON OMOL25

We train a 1B parameter Transformer on the OMol25 4M training split, using the same total training compute budget (measured in FLOPs) as eSEN—a state-of-the-art 6M-parameter equivariant GNN model which serves as our point of reference. The compute budget includes both pre-training (10 epochs) and fine-tuning (60 epochs).[1]

We compare the energy and force mean absolute errors (MAE) for our Transformer to those of eSEN, as well as the training and inference speeds, in Table 1. Despite having no built-in geometric priors, the Transformer achieves competitive accuracy with eSEN on energies and forces. It also trains and runs faster in wall-clock time, benefiting from mature software and hardware optimizations for Transformer architectures (Ansel et al., 2024; Jouppi et al., 2017). See Appendix B.2 for more detailed discussion of the speed.

Table 1: **Out-of-distribution composition validation results.** Transformers match the energy and force errors of a state-of-the-art equivariant GNN (Fu et al., 2025; Levine et al., 2025) under the same training computational budget, while achieving faster training and inference in wall clock time. We estimate FLOPs through the FairChem repository for eSEN and through HuggingFace tooling for our Transformer. We measure the training speed on a single node of 4 H100s and the forward latency on a single A6000 with a system of 100 atoms.

| Model | FLOPs | Forward Latency (ms) | Training Speed (atoms/sec) | Energy MAE (meV) | Forces MAE (meV/Å) |
|---|---|---|---|---|---|
| eSEN-sm-d 6M | $O(10^{20})$ | 26.3 | 32k+ | 129.77 | 13.01 |
| Transformer 1B (Ours) | $8.5 \times 10^{19}$ | 17.2 | 42k+ | 117.99 | 18.35 |

**Further Evaluations.** We evaluate whether Transformers learn rotational equivariance from data alone by measuring the cosine similarity between forces predicted in different rotational frames: $\mathrm{cossim}(\mathbf{R}\mathbf{F}(\mathbf{r}), \mathbf{F}(\mathbf{R}\mathbf{r}))$, where $R$ is a rotation matrix and $\mathbf{F}(r)$ are the model's predicted forces for a system with atomic positions $\boldsymbol{r}$. Averaged over the OMol25 validation set, the similarity exceeds 0.99, consistent with prior findings (Qu & Krishnapriyan, 2024; Neumann et al., 2024; Eissler et al., 2025) that models without explicit equivariance can learn approximate equivariance directly from training data. This value can essentially arbitrarily increase through frame averaging (Puny et al., 2022; Duval et al., 2023) though at the cost of slower inference.

We also test the model in molecular dynamics (MD) simulations. We find that the Transformer can run stable NVT simulations which accurately estimate thermodynamic observables (see Figure 6),

---

[1]FLOPs for eSEN are estimated using the FairChem repository; FLOPs for our Transformer are estimated via HuggingFace tooling.

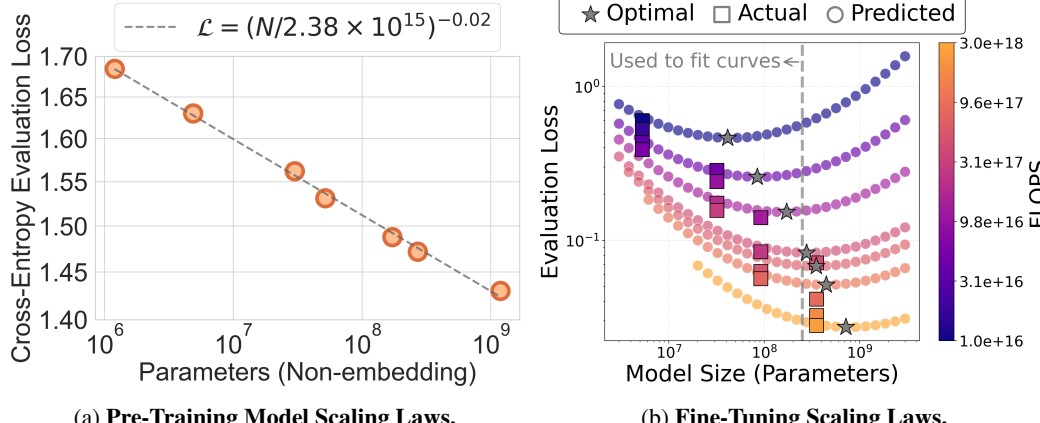

(a) **Pre-Training Model Scaling Laws.**          (b) **Fine-Tuning Scaling Laws.**

Figure 3: **Transformers scale predictably with training resources when modeling molecules.**
(a) We pre-train models of varying sizes up to 1B parameters with other training hyperparameters
held fixed. Evaluation performance improves in a clear power-law relationship with model size. (b)
We train models of varying sizes (5M, 30M, 90M, 350M) for differing number of epochs (1,2,4,6)
using our fine-tuning setup. We fit scaling laws with the three smaller models and make predictions
about the performance of other model sizes trained for varying numbers of epochs. We plot predicted
IsoFLOP curves, where smaller models on each curve are trained for more epochs and larger ones
are trained for fewer epochs. Predicted IsoFLOP curves have a parabolic shape with the optimal
model size and performance for each flop budget following a consistent power-law relationship, in
line with previous work (Hoffmann et al., 2022). The isoflop curves accurately extrapolate to predict
the performance of the larger 350M parameter model.

and it can conserve energy in NVE simulations when fine-tuned to predict a conservative force field:
$\mathbf{F} = -\nabla U$ (see Figure 7). While more rigorous evaluation would be needed before deployment in
large-scale scientific discovery, these results demonstrate that a graph-free Transformer can already
serve as a molecular force field—making it a compelling case study for the representation-learning
analysis in Section 4.3.

## 4.2 SCALING ANALYSIS

Scaling laws describe how model performance changes predictably with training resources. They are
widely used in other areas of machine learning to guide model design and training with predictable
results (Hoffmann et al., 2022; Kaplan et al., 2020; Brown et al., 2020; et. al., 2024; Snell et al., 2024).
If similar laws hold for molecular modeling, they could provide a principled recipe for building larger,
more capable models without long trial-and-error experimentation.

**Pre-Training Scaling Laws.** We train seven Transformers of varying sizes up to one billion param-
eters on the OMol25 4M training split (Levine et al., 2025). All runs use identical hyperparameters,
with only the parameter count varied. We train models for 10 epochs with rotation augmentation,
processing over 2B tokens, comparable to the ten billion token dataset used by Kaplan et al. (2020).
We report detailed hyperparameters and model sizes in Table 2 and Table 3, respectively.

Following prior work (Kaplan et al., 2020; Hoffmann et al., 2022), we assume a power-law relationship
between the cross-entropy test loss $\mathcal{L}$ and model size $N$: $\mathcal{L}(N) = \left(\frac{N}{N_c}\right)^{\alpha}$, where $N$ represents the
number of non-embedding parameters. Figure 3a shows that Transformers continue to predictably
improve in performance with model scale, with no sign of saturation up to 1B parameters.

**Fine-Tuning Scaling Laws.** We next examine scaling during fine-tuning. We train three model
sizes (5M, 30M, 90M) for varying numbers of epochs (1, 2, 4, 6) on OMol25 4M. Using these
results, we fit power-law scaling curves that predict model performance from both model size and
number of training epochs. From these runs, we generate IsoFLOP curves—performance curves
for constant training compute (total floating-point operations). On each curve, smaller models are
trained longer, while larger models are trained for fewer epochs. The predicted IsoFLOP curves

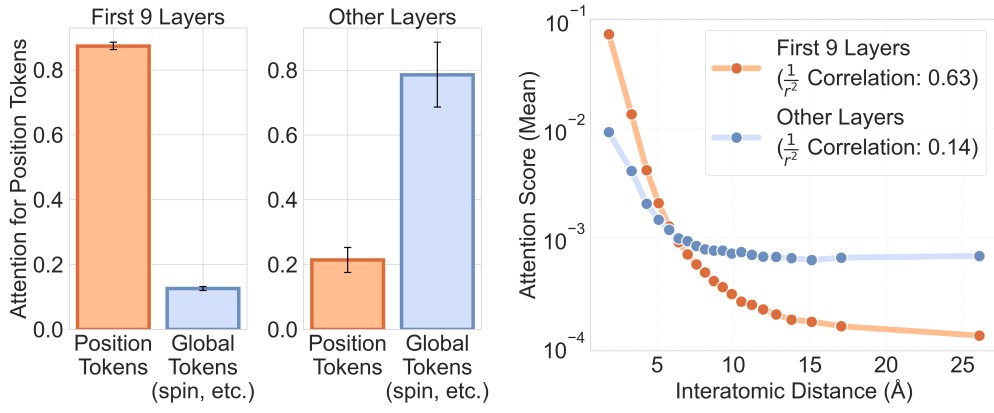

(a) **Attention Distribution by Token Type.**    (b) **Attention versus interatomic distance.**

Figure 4: **Transformers effectively capture local features in early layers and global features in later layers.** (a) We show what fraction of attention from position tokens goes towards other position tokens versus global tokens, such as charge and spin. In the first nine layers, position tokens predominantly focus on other position tokens. In the later layers, attention shifts towards global tokens. (b) We plot attention scores for position tokens against interatomic distance, averaged across validation examples in OMol25. Each dot is the mean attention score within an interatomic distance quantile. Attention in the first nine layers is strongly inversely related to distance. In later layers, position tokens increase their attention to global tokens (e.g., spin and charge) while still allocating some attention to distant atoms. These results suggest that Transformers captures local features in the early layers and then aggregate global information in the final layers.

are parabolic, with an optimal model size for each compute budget that follows a clear power-law relationship, consistent with findings in other domains (Hoffmann et al., 2022). The scaling laws accurately predict the performance of a larger 306M parameter model, which was not used in fitting the curves (see Figure 3b).

**Observations.**    These results show that Transformers for molecular modeling scale predictably with training resources in both pre-training and fine-tuning. Given that performance has not saturated at 1B parameters, and that scaling laws in other fields hold up to much larger model sizes (Touvron et al., 2023b; Brown et al., 2020; Siméoni et al., 2025), it is plausible that accuracy could continue to improve well into the hundreds of billions of parameters. While scaling laws can break down in certain modalities (Henighan et al., 2020), our findings suggest that molecular Transformers may follow the same predictable trends that have driven progress in other fields of ML.

### 4.3 Investigating the Learned Representations of the Transformer

With a Transformer model that demonstrates competitive performance on the OMol25 dataset, we next examine the representations it learns in the absence of graph priors. We focus on the learned **attention scores**—the softmax-normalized dot products between queries and keys—which reveal how the Transformer allocates attention across different layers.

**How does the Transformer Distribute its Attention across Layers?**    To understand how the Transformer distributes its attention, we first analyze the attention score distribution based on token type. For example, we assess how much attention position tokens allocate to each other versus to global tokens, such as those representing charge or spin. We find a clear difference in attention patterns between early layers of the network and later ones. In the early layers of the model (the first nine layers), position tokens devote over $80\%$ of their attention to other position tokens. This pattern shifts in later layers, where attention increasingly focuses on global tokens carrying information such as charge and spin. This change is illustrated in Figure 4a and Figure 10, which compare attention scores for position and global tokens across layers.

The significant change in attention patterns among position tokens throughout the network leads us to investigate their interactions more closely. When we plot the mean attention score against interatomic distance (see Figure 4b), we observe a clear inverse correlation in the first nine layers: as distance

increases, the attention score decreases. In contrast, in the later layers, attention initially decays with distance but then remains roughly constant after around $\sim 12$ Å (Figure 8).

One interpretation of these attention maps is that the Transformer learns to first perform local feature extraction before shifting to global aggregation. In early layers, attention concentrates on nearby atoms, evident from the inverse relationship between attention score and distance (shown in Figure 4b) and the high attention between position tokens (shown in Figure 4a). Interestingly, this distance-dependent attention drops off around the 6–12 Å range, coinciding with the radius cutoffs commonly used in traditional graph-based MLIPs (Kovács et al., 2023; Fu et al., 2025; Levine et al., 2025). These findings suggest that the Transformer naturally learns to extract local features in the early layers, without relying on predefined graph construction algorithms and message-passing schemes. In later layers, the model aggregates global information about the molecule, evidenced by the relatively constant attention scores at larger distances and increased focus on global tokens. This global attention could allow the model to refine its representations with long-range interactions and global molecular properties like charge and spin.

**Investigating Adaptive Attention Patterns.** We have observed a clear pattern of attention decay with distance, which the Transformer learns naturally from data without relying on a predefined graph structure. This prompts us to explore whether this decay pattern is adaptive. In contrast, GNNs often use a hard-coded cutoff radius to designate which pairs of atoms are considered neighbors. However, this fixed radius may be optimal for one molecule but not for another (Kreiman & Krishnapriyan, 2025; Giovanni et al., 2023; Dwivedi et al., 2023). Furthermore, different radii might even be warranted for different atoms *within the same molecule*. For example, if an atom is in a tightly packed region with many neighboring atoms close by, short-range interactions might overwhelm long-range interactions. However, if the atom is isolated from the rest of the molecule and is distant from most other atoms, then long-range interactions would increase in relevance.

To investigate whether the Transformer can learn these more flexible attention patterns, we define the *effective attention radius* $R_i$ for atom $i$ as follows:

$$R_i = \inf\left\{ R \in \mathbb{R}_{\geq 0} : \sum_{\substack{j: \\ \|\boldsymbol{r}_i - \boldsymbol{r}_j\|_2 \leq R}} a_{ij} \geq \delta \right\}, \tag{1}$$

where $\boldsymbol{r}_i$ is the position of atom $i$ and $a_{ij}$ is the attention score between atom $i$ and $j$. This radius is the smallest distance that contains $\delta = 90\%$ of atom $i$'s total attention. Next, we use the median neighbor distance as a proxy for how isolated or densely packed an atom is. For layers 1-9, which exhibit the strongest attention decay with distance, we plot the mean effective attention radius against the median neighbor distance. We observe a clear positive trend: as the median neighbor distance increases, the effective attention radius also increases. This suggests that the model can adapt its effective attention radius per atom based on the local atomic environment, such as how tightly packed the surrounding region is (see Figure 5 and Figure 9).

Finally, we observe that specific attention heads learn other varied flexible attention patterns that are difficult to anticipate or hard-code *a priori*. For example, we observe certain heads exhibit a $k-$nearest-neighbor behavior, where attention is directed based on the rank of neighboring atoms rather than simply decreasing with distance (see Figure 11). Other heads show non-monotonic attention patterns, and some even increase attention with distance (see Figure 12).

**Observations.** The analysis in this section shows some of the advantages of relaxing inductive biases in model design. Traditional GNNs require predetermined cutoffs or neighbor definitions, which risks underfitting long-range interactions or overfitting local structures. This rigidity can result in suboptimal performance across different molecule topologies (Dwivedi et al., 2023; Kreiman & Krishnapriyan, 2025; Giovanni et al., 2023). In contrast, Transformers dynamically adapt their receptive field for each atom, expanding in sparse regions and contracting in dense ones. They can also support a variety of diverse head-specific behaviors such as $k$-nearest-neighbors (where attention depends on neighbor rank instead of distance), non-monotonic attention patterns, or even attention that increases with distance. This adaptivity enables Transformers to leverage the advantages of graph-based inductive biases without the rigidity of manual design.

## 5 DISCUSSION AND CONCLUSIONS

We investigated the learned representations of an unmodified, graph-free Transformer trained on chemical data. As a starting point for our investigation, we found that an appropriately trained Transformer that uses no physical inductive biases can achieve competitive molecular force and energy errors on the OMol25 dataset using the same computational budget as a state-of-the-art GNN. We found that the Transformer predictably improves in performance with scale, in line with previous literature in other fields of ML. Finally, we explored the attention maps of our Transformer and found that it naturally learns physically consistent behaviors that are hard-coded in GNNs. Importantly, since the Transformer includes no explicit graph, we found that it exhibits adaptive patterns—such as an effective radius cutoff that varies based on atomic environments—which would be hard to specify *a priori* in a traditional GNN.

**Limitations.** While our findings demonstrate that Transformers can accurately approximate energies and forces on the OMol25 dataset, it is important to acknowledge that fully unconstrained models may have issues adhering to certain physical principles. While this presents a challenge, evidence from many other fields of ML, including those that also have physical principles (Team et al., 2024; et. al., 2024; Kim et al., 2024), and new MLIP architectures (Qu & Krishnapriyan, 2024; Neumann et al., 2024) suggest that symmetries can be learned directly from data with im-

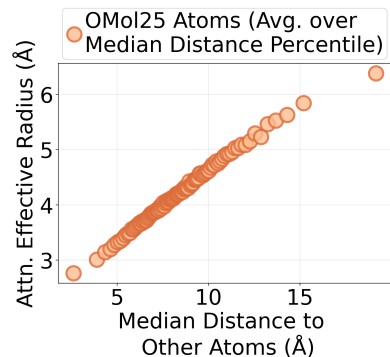

Figure 5: **Relationship between attention effective radius and atom density.** Averaging over atoms in the OMol25 validation set, we plot the effective attention radius versus the median distance to other atoms. Each dot is the mean effective radius within a median neighbor distance percentile. We define the effective radius as the minimum distance within which 90% of an atom's attention mass is concentrated (see Equation (1)). The model learns to *adaptively* increase its effective attention radius when an atom is more isolated, and to decrease it when atoms are tightly packed.

proved training strategies and an expressive model. We think it is an interesting direction for future work to examine if physical laws can be taught to unconstrained models with improved training strategies. In cases where strict adherence to physical constraints is necessary, one potential approach is to fine-tune on top of the Transformer representations to improve the performance of a traditional MLIP. Distillation methods could also be used to leverage the knowledge of a more general model when training a smaller, specialized MLIP with physical constraints (Amin et al., 2025). Since our Transformers can operate on continuous inputs, the model could also be fine-tuned to predict forces as an energy gradient (see Appendix A), and other constraints could be imposed after pre-training.

**Future Work.** The flexibility of Transformers reveals several advantages not present in current MLIPs. Since the input is simply represented as a string of tokens, it is straightforward to use new input formats, including, but not limited to, multi-modal experimental data and conditioning on the level of DFT theory. Using a discrete output head also provides a simple version of uncertainty quantification (Liu et al., 2024; Kuhn et al., 2023; Abdar et al., 2021) (see Appendix A.3). Finally, since Transformers learn the full joint distribution over positions, forces, and energies, Transformers could be used both as a force field and as a generative model (Arts et al., 2023) of atomic structures.

More generally, insights from previous deep learning research suggest that when enough data is available, expressive models that leverage powerful optimization algorithms on modern hardware can outperform methods which rely on hand-crafted inductive biases (Brown et al., 2020; Vaswani et al., 2023; Dosovitskiy et al., 2021; Kim et al., 2024). While some inductive biases may be beneficial for narrow problem settings, tackling new problems often requires designing new biases for each task. In contrast, general search and learning methods can discover inductive biases directly from data, and perhaps even discover more flexible solutions that are hard to anticipate *a priori* (Sutton, 2019).

Our findings suggest that Transformers appear capable of learning many of the graph-based inductive biases typically incorporated in current ML models for chemistry. We hope these findings point towards a standardized, widely applicable architecture for molecular modeling that draws on insights from the broader deep learning community.

**Reproducibility Statement.** We describe our experimental setup throughout Section 3 and Section 4. We also provide more detailed descriptions, exact hyperparameters, and computational usage in Appendix A, Appendix B, and Appendix C. We will release our code publicly.

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

## A    FURTHER EXPERIMENTS

We take first steps at exploring the downstream utility of Transformers for molecular modeling. We run MD simulations with our Transformer in Appendix A.1 before exploring the representations in more detail in Appendix A.2. Since Transformers learn the joint distribution over positions, forces, and energies, we then examine in Appendix A.3 whether Transformers can be used for uncertainty quantification to identify where current MLIPs make mistakes.

### A.1    MOLECULAR DYNAMICS SIMULATIONS ON OMOL

To further evaluate the Transformer beyond energy and force errors, we run MD simulations. We first run NVT simulations and calculate a thermodynamic observable (the distribution of interatomic distances) from the simulations to evaluate the quality of the dynamics. The distribution of interatomic distances is a commonly used observable which characterizes 3D molecular structures (Zhang et al., 2018; Fu et al., 2023; Raja et al., 2025) and is defined as:

$$h(r) = \frac{1}{n(n-1)} \sum_{i=1}^{n} \sum_{i \neq j} \delta(r - ||\mathbf{X}_i - \mathbf{X}_j||), \tag{2}$$

where $\mathbf{X} \in \mathbb{R}^{n \times 3}$ are the positions of the $n$ atoms of a molecule and $\delta$ is the Dirac Delta function. For our evaluation metric, we calculate the $h(r)$ MAE with respect to a reference:

$$\int_{r=0}^{\infty} dr |\langle h^*(r) \rangle - \langle \hat{h}(r) \rangle|, \tag{3}$$

where $\langle \cdot \rangle$ represents an average over structures sampled from the predicted ($\hat{h}(r)$) or reference ($h^*(r)$) Boltzmann distribution.

We simulate 10 random validation molecules for 100ps using a 0.5 fs timestep. We use a Langevin thermostat at 500K with a friction of $0.01$ fs$^{-1}$. We also run simulations with the eSEN-sm models (both conserving and direct prediction versions) (Levine et al., 2025). We use simulations from the UMA-S model (Wood et al., 2025) as a reference since it is a larger model trained on significantly more data. The Transformer has a $h(r)$ MAE (see Equation (3)) of 0.040 relative to UMA-S for the 10 molecules (compared to 0.077 and 0.065 for eSEN-sm-d and eSEN-sm-c, respectively), showing that Transformers can be applied to run molecular dynamics (see Figure 6).

Since conservative force fields are important for many downstream applications, we also explore fine-tuning our Transformer to predict forces as the gradient of the energy. We selected the first 5 molecules used to evaluate the NVT simulations and additionally ran NVE simulations for 100ps using a 0.5fs timestep. The fine-tuned Transformer was able to accurately conserve energy, where as a direct-prediction model experiences significant energy drift (see Figure 7). While we observed that fine-tuning with energy gradients made training more unstable, this provides a proof of concept that graph-free Transformers can still be accurately used for downstream molecular tasks. We think it is an interesting direction for future work to examine how other physical constraints can be taught to Transformers after the initial training phase.

### A.2    ANALYSIS OF LEARNED REPRESENTATIONS OF GRAPH-FREE TRANSFORMERS

We provide more detailed plots for our attention score analysis in Section 4. We breakdown by layer in Figure 8 the relationship between attention score and interatomic distance. We observe a clear positive trend of the effective attention radius increasing with the median neighbor distance across different layers in Figure 9. Importantly, each layer is able to flexibly learn its attention pattern, without having to rely on a predefined graph.

We provide a more detailed breakdown of the attention distribution by token type. We group the input tokens into four semantic buckets of **Positions**, **Charge**, **Spin**, and **Delimiter**. The first three buckets encode the atomic positions, the molecular charge, and the molecular spin, respectively. The last bucket includes all other tokens, which encode delimiter information of where each section begins and ends (e.g., [POS], [POS_END] tokens). We then produce four figures, each restricting rows of the attention matrix to each of the four buckets, respectively. For each figure, we plot the evolution of the

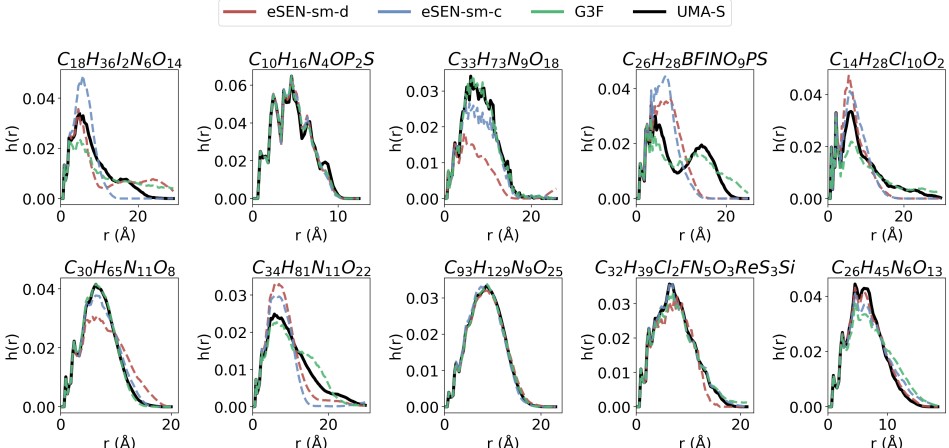

Figure 6: **Transformers can accurately run MD simulations relative to models with strong inductive biases.** We select 10 random molecules from the OMol validation set and run NVT MD simulations at 500K for 100ps using a 0.5 fs timestep. We plot the estimated distribution of interatomic distances $h(r)$ for the eSEN-sm models (both direct and conserving versions) and the UMA-s model as reference. Transformers accurately reproduce the distribution of interatomic distances relative to these models without using any physical inductive biases.

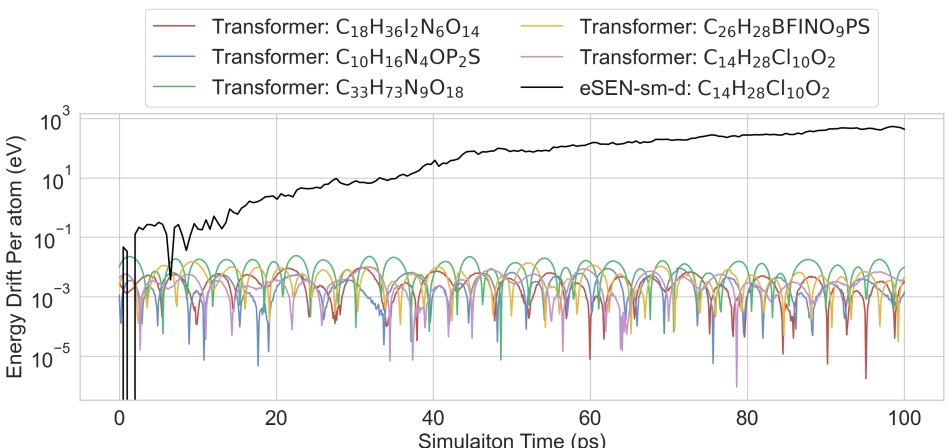

Figure 7: **Transformers can conserve energy during NVE simulations when fine-tuned to predict conservative forces.**

Figure 8: **Attention is strongly inversely correlated with distance in layers** 1-9**.** Attention decays steeply with increasing interatomic distance in layers 1-9. This implies that the model is learning to attend predominantly to local interactions in early layers.

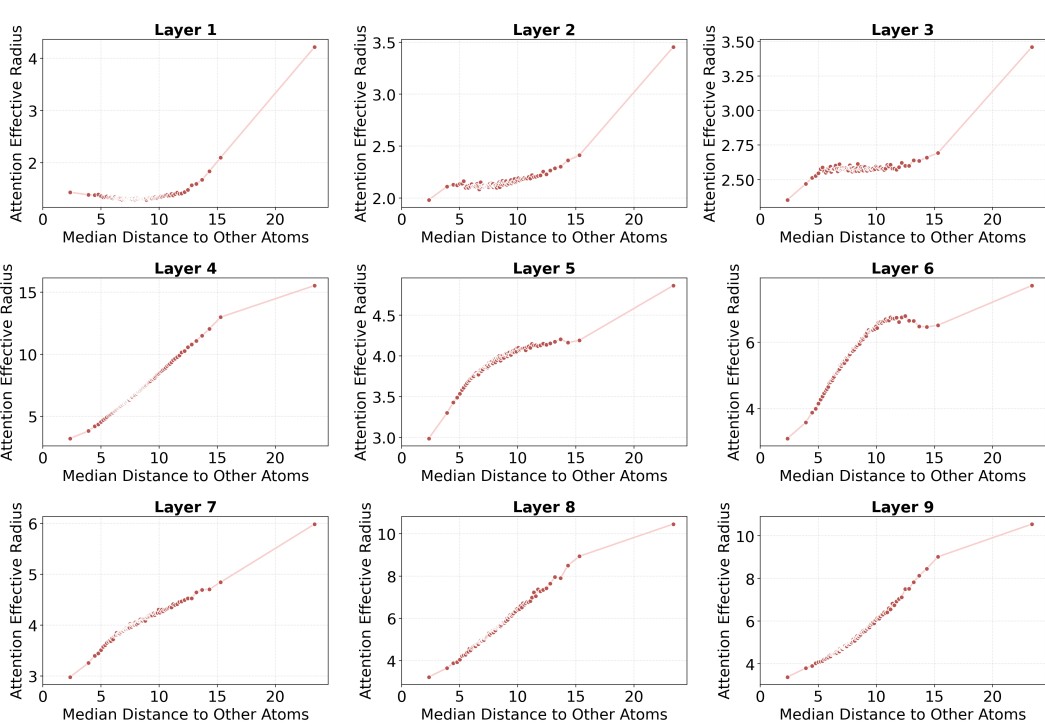

Figure 9: **Effective attention radius is adaptive to how tightly packed an atom is within a molecule.** We observe a clear positive trend between effective attention radius and median distance to neighbors. Within certain layers (e.g., layer 4), radius can go as high as 15Å and as low as 2Å depending on the molecule and atom within that molecule.

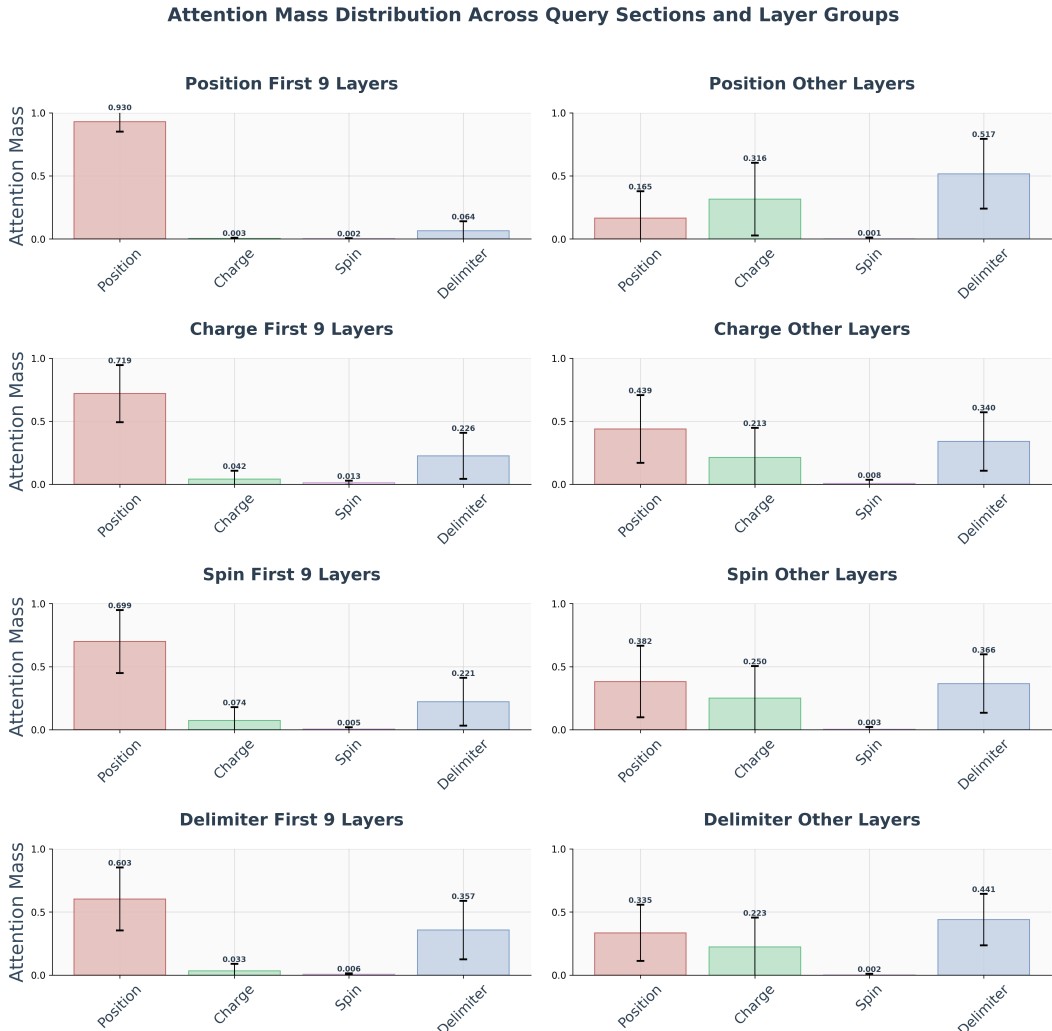

Figure 10: **Interatomic attention dynamics dominate in early layers while global attention dominates in later layers.** In the first 9 layers, position tokens attend almost exclusively to the other position tokens. They then shift attention to other input tokens in later layers, suggesting that they're accessing global graph information. We observe that barely any attention is paid to the spin tokens. We attribute this to the fact that high-spin molecules are rare in the training dataset (see Figure 14), suggesting that the model needs more varied spin data to learn how to use spin effectively.

attention mass distribution across each of the four buckets, as we move from the first 9 layers to later layers (see Figure 10). In the first 9 layers, **Positions** tokens attend almost exclusively to the other **Positions** tokens. They then shift attention to other input tokens in later layers. This is consistent with the picture observed earlier that in layers 1-9 interatomic attention dynamics dominate, while it dies down in later layers. It is natural to ask what relevant information these non-position tokens carry, in addition to charge and spin information, by the time **Positions** tokens shift attention to them. To this end, if we take a look at the mass distribution of **Charge**, **Spin**, and **Delimiter** tokens (last three rows of Figure 10), we notice that all of them them predominantly ($> 60\%$) attend to **Positions** tokens in the first 9 layers. This suggests non-position tokens aggregate global graph information in layers 1-9, acting as information banks. **Positions** tokens, after they shift focus from interatomic attention, then access this global information in later layers by attending to them. We acknowledge that it appears that virtually no attention is given to **Spin** tokens. We attribute this to the fact that high-spin molecules are exceedingly rare in the training dataset (see Figure 14), suggesting that the model needs more exposure to learn how to use spin effectively.

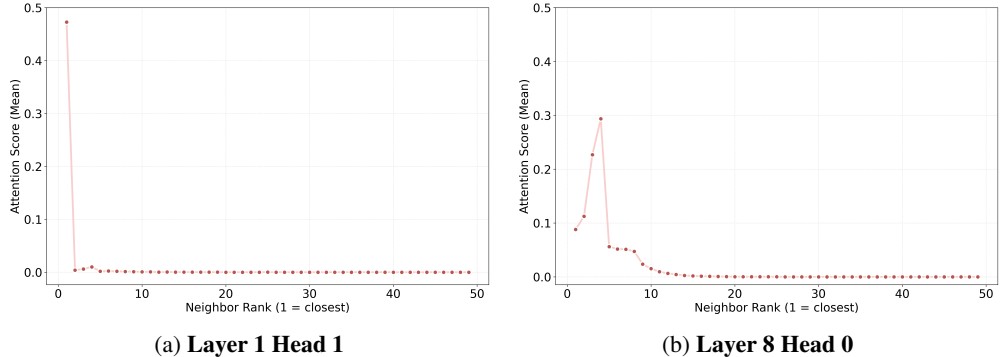

(a) **Layer 1 Head 1**  (b) **Layer 8 Head 0**

Figure 11: **Certain heads in certain layers exhibit unique rank-based attention behavior.** (a) Layer 1 head 1 exhibits a 1-nearest-neighbor pattern where attention is overwhelmingly placed on the closest neighbor and almost no attention beyond. (b) Layer 8 head 0 increases attention with rank and peaks at rank 4, before it drops sharply, and decreases gradually after.

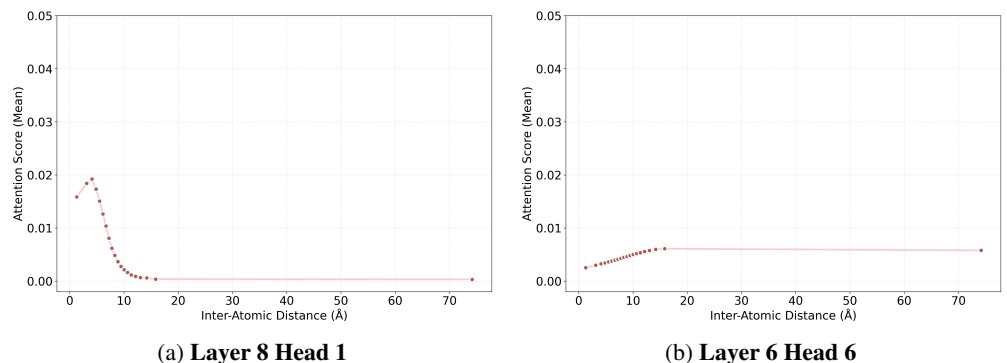

(a) **Layer 8 Head 1**  (b) **Layer 6 Head 6**

Figure 12: **Certain heads in certain layers exhibit unique attention relationships with distance.** (a) Layer 8 head 1 exhibits non-monotonic attention that increases at small distances, peaks at $\sim 5$Å, then decays gradually. (b) Layer 6 head 6 exhibits positive correlation of attention with distance, where attention increases steadily putting more mass on far-away atoms.

Finally, we end this section by observing that certain heads in certain layers exhibit unique interatomic attention behavior that deviates from the smooth monotonic decay of attention with distance we observe when we average over heads. Such patterns are evident when we plot against neighbor absolute distance, as well as rank. We note two examples that stand out in each domain. In Figure 11a, layer 1 head 1 exhibits a 1-nearest-neighbor pattern where attention is overwhelmingly placed on the closest neighbor and almost no attention beyond. And in Figure 11b, layer 8 head 0 increases attention with rank, peaks at rank 4, before it drops sharply, and decreases gradually after. In Figure 12a, layer 8 head 1 exhibits non-monotonic attention that increases at small distances, peaks at $\sim 5$Å, then decays gradually. In Figure 12b, Layer 6 head 6 exhibits positive correlation of attention with distance, where attention increases steadily, putting more mass on distant atoms. These examples demonstrate non-monotonic and long-range interatomic attention patterns, which appear to be useful for molecular energy and force prediction, but are difficult to anticipate or hard-code *a priori*.

### A.3 UNCERTAINTY QUANTIFICATION

Since Transformers learns the whole joint distribution over positions, forces, and energies, our model can estimate the log probability of sequences. By looking at the log probability of an atomic structure predicted by our model, Transformers can be used as a tool for uncertainty quantification to identify structures that are out-of-distribution relative to the training dataset (Liu et al., 2024; Kuhn et al., 2023; Abdar et al., 2021).

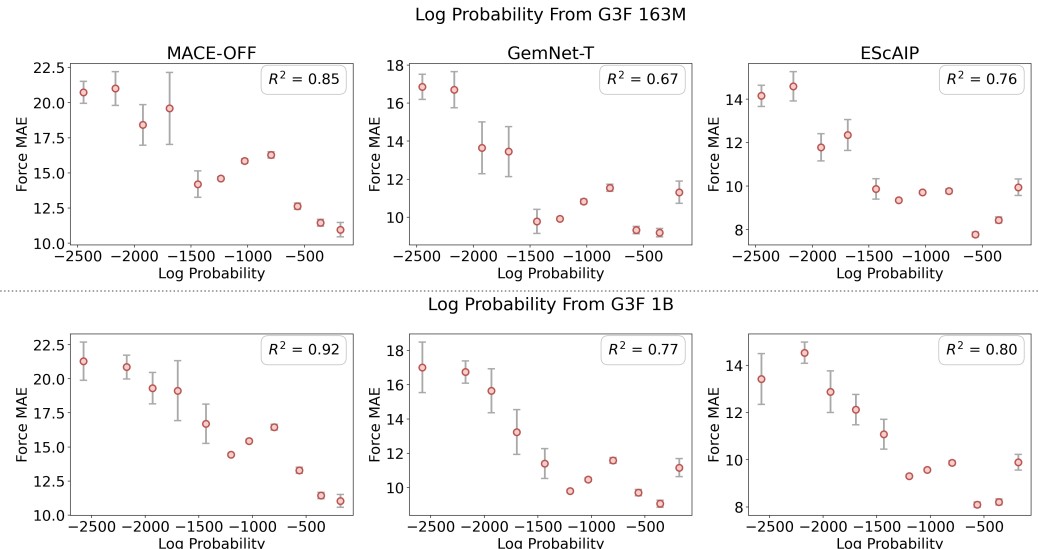

Figure 13: **Transformers can be used for uncertainty quantification for other MLIPs.** Since Transformers learns a joint distribution over positions, forces, and energies, it can compute log probabilities of new structures encountered at test-time. A higher log probability corresponds to the structure being closer to the training data. Transformers's predicted log probabilities correlate with force errors for other models like MACE-OFF, GemNet, and EScAIP. The correlations is stronger as Transformers is scaled up from 163M (top) to 1B parameters (bottom). Force errors are in meV/Å.

We use the SPICE dataset as a case study (Eastman et al., 2023) and pre-train two Transformers (163M and 1B). We compare the predicted log probabilities from the two models to the force errors of commonly used MLIPs on the SPICE dataset (Gasteiger et al., 2021; Kovács et al., 2023; Qu & Krishnapriyan, 2024).

Figure 13 shows that the Transformers' log probabilities are highly correlated with the force errors of MACE-OFF (Kovács et al., 2023), GemNet-T (Gasteiger et al., 2021), and EScAIP (Qu & Krishnapriyan, 2024), indicating that it is accurately capturing the training distribution. As we scale the Transformer, it is able to better predict errors across all models. Although this is just a proof-of-concept, this previews how the flexibility of the Transformer can be used to easily tackle a broader class of problems, beyond just energy and force prediction.

# B  EXPERIMENT DETAILS

## B.1  TRAINING

**Pre-Training.** We follow previous scaling laws literature when choosing hyperparameters for our pre-training experiments in Section 3 (Hoffmann et al., 2022; Kaplan et al., 2020). We use rotation augmentation during pre-training. We report our exact hyperparameters in Table 2 and our model sizes in Table 3.

We ran an ablation on the SPICE dataset (due to computational constraints) to evaluate the utility of adding the continuous input. With only the discretized input, the a 250M G3F model trained on the SPICE dataset has a force MAE of $\sim 200$meV/Å after 5 epochs of training, whereas adding the continuous input brings this down to $\sim 60$meV/Å. After the full pre-training, the fully discrete model has force errors of $\sim 60$meV/Å versus $\sim 40$mev/Å for the model with the continuous embeddings.

We note that the continuous values present in molecular datasets are heavy tailed and span orders of magnitude (see Figure 14). This causes large discretization errors at the tails with our quantile binning method. Giving the continuous sequence to the model as an additional input ameliorates this

issue (see Section 3 and Figure 1). We think it is an interesting direction for future work to examine other discretization schemes for molecules.

**Fine-tuning.** We also largely followed the same hyperparameters for fine-tuning; however, we found training to be more unstable during fine-tuning. Following previous large scale training recipes (Touvron et al., 2023b; Brown et al., 2020), we would resume training following an instability after halving the learning rate. We found it important to use a large enough batch size ($> 1024$ structures) and clip the gradient norms around 100—not too high so as to destabilize training, but not too low so as to kill progress and get stuck in local minima. We also found it important to not use a discrete embedding for charge and spin, rather to treat it as a continuous signal to stabilize training, enabling gradients to propagate for these embeddings for any training sample. We hypothesize that this stabalized training due to the imbalance of extreme charge and spin in the dataset (see Figure 14). We also use rotation augmentation during fine-tuning. We provide detailed hyperparameters in Table 2.

### B.2 WHY IS THE TRANSFORMER FASTER DESPITE HAVING SO MANY MORE PARAMETERS?

There are many ways to compare the efficiency of models. Comparisons can be made between model parameters, model FLOPs, or wall-clock time. These can each be misleading for their own reasons, and we provided measures of each in Table 1. Raw parameters can be misleading since parameters can be used multiple times during a model's forward pass. For example, GNNs often use parameters multiple times per node and edge to construct messages in the message passing step. FLOPs alone can be misleading when comparing different model types since different types of operations can be implemented at different speeds on modern hardware. For example, a sequential operation could be slower compared to a parallel one even if they have the same number of FLOPs, and sparse operations (like those in GNNs) are often slower to implement then dense ones (like those in Transformers). Finally, wall-clock time is system dependent and can improve with the next generation of hardware. Regardless, we find that Transformers leverage mature software and hardware frameworks to run efficiently, even compared to GNNs with far fewer parameters.

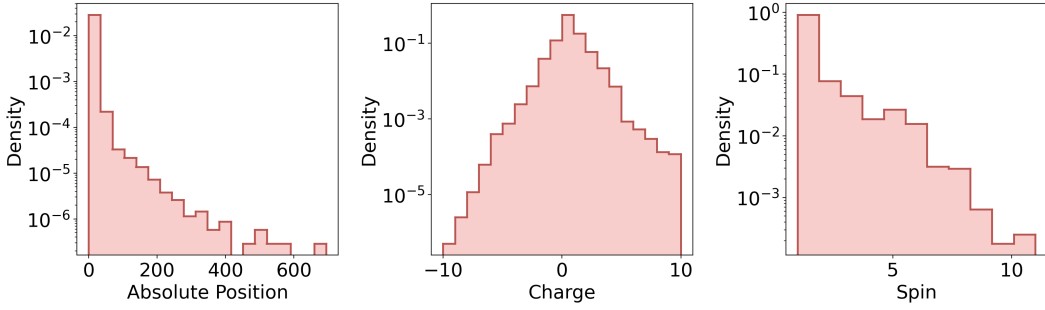

Figure 14: **Distribution of input features in the OMol dataset.** The continuous values in molecular datasets span multiple orders of magnitude and are heavy tailed.

## C  COMPUTATIONAL DETAILS

We trained our larger models for the scaling experiments on a cluster of A100 and V100 GPUs. Training the 1B parameter model took $\sim 750$ A100 hours using only data parallelism. We used gradient accumulation to achieve the effective batch size reported in Table 2. We trained the 350M, 250M, and 170M parameter models for $\sim 600$ V100 hours. The smaller models were trained on a single A6000 for up to 100 hours. We used compilation (Ansel et al., 2024) to speed up training. We think it is an interesting direction for future work to explore how systems innovations for the Transformer architecture can further speed up training and inference (Dao et al., 2022; Ansel et al., 2024; Kwon et al., 2023).

```
<BOS>

[POS]
a_35:  <NUM_579>
a_35:  <NUM_728>
a_6:  <NUM_657>
a_1:  <NUM_456>
a_1:  <NUM_766>
a_8:  <NUM_240>
a_1:  <NUM_231>
a_1:  <NUM_340>
[POS_END]

[TARGET]  <NUM_target_125>  [TARGET_END]

[FORCE]
<NUM_force_214>  <NUM_force_35>  <NUM_force_42>
<NUM_force_75>  <NUM_force_227>  <NUM_force_72>
<NUM_force_49>  <NUM_force_210>  <NUM_force_87>
<NUM_force_209>  <NUM_force_50>  <NUM_force_197>
<NUM_force_64>  <NUM_force_94>  <NUM_force_219>
<NUM_force_229>  <NUM_force_58>  <NUM_force_210>
<NUM_force_137>  <NUM_force_206>  <NUM_force_30>
<NUM_force_25>  <NUM_force_107>  <NUM_force_174>
[FORCE_END]

<EOS>
```

Figure 15: **Example Discretized Input for the Model.** While we presented a simplified version of the input in the main text for clarity in Figure 2 and Figure 1, we show a full example of a discretized input string here. We include special tokens to tell the model when to start predicting forces and energies (denoted by target). a_i represents atomic number $i$.

Table 2: **Hyperparameters for training Transformers.**

| Hyperparameter | Pre-Training Value | Fine-Tuning Value |
|---|---|---|
| Learning Rate | $3 \times 10^{-4}$ | $3 \times 10^{-4}$ |
| Weight Decay | 0.0 | $1 \times 10^{-3}$ |
| Optimizer | Adam | Adam |
| Epochs | 10 | 60 |
| Batch Size | 1024 | 2048 |
| Warmup Percentage | 5% | 10% |
| LR Scheduler | Cosine Decay | Cosine Decay |
| Num Bins Force | 4096 | - |
| Num Bins Target | 2048 | - |
| Num Bins Pos (joint embedding) | $10^3$ | - |
| Num Bins Pos (1D discretization) | 512 | - |
| Output head(s) | Linear readout to logits | Energy + Force Gated MLPs |
| Attention Mask | Causal | Bi-directional |
| Loss | Cross-entropy | MAE (only E+F) |
| Clip Grad Norm | 1.0 | 100 |

## D    BROADER IMPACT

We are aware that computational chemistry methods can be used to create and study both good and bad chemical systems. Our work is meant to advance the general field of MLIPs, and we are

Table 3: **Model sizes for scaling experiments.** Model hyperparameters were adapted from Hoffmann et al. (2022).

| Number of Non-Embedding Parameters | Hidden Dimension | Num Layers | Intermediate Size | Num Heads |
|---|---|---|---|---|
| 800k | 100 | 3 | 400 | 4 |
| 5M | 256 | 4 | 1024 | 4 |
| 30M | 576 | 5 | 2304 | 9 |
| 50M | 576 | 9 | 2304 | 9 |
| 90M | 640 | 13 | 2560 | 10 |
| 170M | 768 | 18 | 3072 | 12 |
| 250M | 1024 | 16 | 4096 | 16 |
| 350M | 1024 | 20 | 4096 | 16 |
| 1.2B | 1792 | 23 | 7168 | 14 |

cognizant of the ethical implications of our work and conduct our research in a responsible manner. We will release our models in a responsible manner, and provide detailed instructions on how to use them.

