# OpenReview forum: "Transformers Adaptively Learn Molecular Structures Without Graph Priors"
_ICLR.cc/2026/Conference — Submitted to ICLR 2026_

### Official Review · Reviewer_RHEQ · 2025-10-30

**Soundness:** 3
**Presentation:** 3
**Contribution:** 2
**Rating:** 2
**Confidence:** 4

**Summary:**

This paper tokenizes all the attributes of molecules and employs a vanilla Transformer to capture features. The authors assert that this approach does not rely on physical priors. Experiments demonstrate that many favorable properties of GNNs can emerge in this Transformer, achieving results comparable to state-of-the-art GNNs.

**Strengths:**

- This paper is well-written.

**Weaknesses:**

1.The main argument of this paper lacks innovation. The Transformer learning the representation of GNN is similar to the story of CNN and ViT: when data is scarce, ViT performs poorly, but it performs better when trained with large-scale data. This is because the Transformer is a fully connected model in all dimensions (sequence and embedding dimensions), with a high upper limit of expressive power, but it is prone to overfitting. Whether in vision or molecules, researchers introduce priors to reduce overfitting, thereby reducing training costs and improving generalization ability. Moreover, replicating ViT in the molecular field may not make sense, as existing datasets can only cover a small part of the molecular field. Therefore, the conclusions and results of this paper are likely due to large-scale data training, which has little significance in the field of science.

Graph Neural Networks are originally similar to Transformers (Transformer can be regraded as a fully connected graphs). If you use fully connected graphs for learning, the edge features can also learn the knowledge of cutoff.

2.I am concerned about the generalization of this paper. Although the Transformer has achieved remarkable results in NLP, its effectiveness in continuous space is still in doubt. Methods that divide 3D space into grids may reduce the model's generalization ability. For example, when inferring large protein molecules, GNNs with priors may show more stable results.

The Transformer tends to memorize the patterns of the training set rather than performing inference in many tasks [1], which is fatal for the mathematics in the molecular space.

3.The pure Transformer structure appears to be non-equivariant. Equivariance is a common characteristic of all molecules, so why give up this prior and increase the difficulty of model training? This requires more data or data augmentation. The Equiformer [2] and SE(3)-Transformer [3] models extract features using the Transformer while preserving equivariance, which I believe is a more rational and economical approach.

4.This paper lacks novelty in both algorithms and training. Its biggest contribution is a foundation model for molecules. I believe that such a contribution can only be valuable if it is open-sourced. Unfortunately, I couldn't find the open-sourced model and thus couldn't test it.。

[1] Is Chain-of-Thought Reasoning of LLMs a Mirage? A Data Distribution Lens

[2] EquiformerV2: Improved Equivariant Transformer for Scaling to Higher-Degree Representations

[3] SE(3)-Transformers: 3D Roto-Translation Equivariant Attention Networks

**Questions:**

See Weaknesses.

---

> ### Author Response · Authors · 2025-11-24
>
> We are confused due to a number of comments being unrelated to the contents of our paper. We reply to individual points below:
>
> > The main argument of this paper lacks innovation.
>
> The field of molecular modeling with ML, and especially the field of machine learning interatomic potentials (MLIPs), has largely relied on using GNNs. To our knowledge, there is no prior work that uses an unmodified Transformer as an MLIP. While broader molecular property prediction work has explored varied architectures, some of which take inspiration from the Transformer, many of these works make substantial modifications to the standard Transformer, for example by modifying the attention mechanism or adding graph-based featurization (see our related works for more discussion).
>
> We present a recipe that shows how to train an unmodified Transformer on molecular data, building a bridge between the field of molecular ML (which has relied on more domain specific architectures) to a large part of the deep learning community that has largely converged to the Transformer. The ViT line of work was very successful in this vein, and this work was our version of investigating the role of the different inductive biases available given the current dataset scales in computational chemistry.
>
> > Moreover, replicating ViT in the molecular field may not make sense, as existing datasets can only cover a small part of the molecular field. Therefore, the conclusions and results of this paper are likely due to large-scale data training, which has little significance in the field of science.
>
> “Large-scale data training” is precisely one of the main parts of our paper that we are investigating, and also why modern deep learning has worked. We run our experiments on the recently released OMol25 dataset, which is a massive computational chemistry dataset of over 100M+ molecular structures. Given the scale of these recently released datasets for the MLIP community, we investigate for the MLIP community what is capable at the current data scales. There is an immense effort by the community to continue curating larger datasets, and we find that a model with minimal inductive biases can already achieve competitive energy and force error with the current scientific datasets.
>
> Compared to the internet-scale language and vision data available, our result that data driven approaches also show promise for computational chemistry can inform the future tradeoff between architecture design, data curation, and training recipe refinement.
>
> > I am concerned about the generalization of this paper. Although the Transformer has achieved remarkable results in NLP, its effectiveness in continuous space is still in doubt.
>
> Could the reviewer clarify what this means? Transformers have been successfully used in continuous spaces across vision [2], diffusion models [1], and robotics [3], to name a few application areas.
>
> >  Methods that divide 3D space into grids may reduce the model's generalization ability. For example, when inferring large protein molecules, GNNs with priors may show more stable results.
>
> We note that after fine-tuning, everything is done in a completely continuous space. We also evaluate the models in our work on the OMol25 dataset, the largest and most diverse computational chemistry dataset to date. Please let us know if there is a concrete comparison you think would help make the paper stronger.
>
> > The Transformer tends to memorize the patterns of the training set rather than performing inference in many tasks [1], which is fatal for the mathematics in the molecular space.
>
> Could the reviewer clarify what this means? The reference from the reviewer seems to be about chain-of-thought reasoning; however, we emphasize that we are not doing autoregressive generation with our models since we focus on a bi-directional regression task.
>
> > The pure Transformer structure appears to be non-equivariant. Equivariance is a common characteristic of all molecules, so why give up this prior and increase the difficulty of model training? This requires more data or data augmentation.
>
> We note that we use the same total amount of training data compared to the equivariant GNN (OMol25 4M training split). There is previous successful MLIP work that has already shown that an accurate model can be learned without built-in rotation equivariance [4,5].

---

> > ### Author Response · Authors · 2025-11-24
> >
> > > The Equiformer [2] and SE(3)-Transformer [3] models extract features using the Transformer while preserving equivariance, which I believe is a more rational and economical approach.
> >
> > As discussed in our related works, we note that Equiformer is __not__ a pure Transformer, rather it uses an attention mechanism over a pre-defined graph. The SE(3)-Transformer takes a similar approach, where it uses an attention mechanism over a local neighborhood (we will add this paper to our related works). We explicitly see if an unmodified Transformer can learn the local interactions without a pre-defined graph.
> >
> > While we agree that rotational equivariance is inherent to molecular spaces, we note that there has been success both in vision [2], robotics [3], and MLIPs [4][5] investigating the role of physical inductive biases. Along with [4][5], we find that rotational equivariance can be efficiently learned without the cost of tensor product operations. We are open to working with the reviewer to make the paper stronger and are happy to continue discussing any concrete criticisms or comments that would clarify this comparison.
> >
> > > This paper lacks novelty in both algorithms and training. Its biggest contribution is a foundation model for molecules. I believe that such a contribution can only be valuable if it is open-sourced. Unfortunately, I couldn't find the open-sourced model and thus couldn't test it.
> >
> > As far as we are aware, we are the first to apply an unmodified Transformer as an MLIP, a domain traditionally dominated by equivariant GNNs. Our main contribution is not a specific architecture, but rather a systematic investigation into the role of inductive biases in learning molecular representations and the extent to which data-driven approaches work for MLIPs. Our work identifies and studies a set of key design choices (how structural information is encoded, what learning objectives are used, and which hyperparameters enable stable training and scaling behavior) that collectively form a general **recipe** for training MLIPs with few inductive biases.
> >
> >
> > As far as we are aware, we are the first to apply an unmodified Transformer as an MLIP, a field which is dominated by equivariant GNNs. **Our main contribution is not a specific model itself, rather the investigation into the role of inductive biases for learning molecular representations and the effectiveness of data-driven approaches for MLIPs.** There are a number of design choices (how to input the data, what learning objective to use, which hyperparameters lead to scaling laws and stable training) which together culminate in a general **recipe** that provides evidence that models with few inductive biases can learn meaningful molecular representations with currently available chemical data.
> >
> > This recipe provides evidence that models with minimal inductive biases can learn meaningful molecular representations using currently available chemical data, and it offers a framework for balancing computational cost, data curation, and model development in future MLIP research. This is in the same spirit of previous successful work in other fields of ML [2][3][6]. We regardless plan to release the code and models upon publication.
> >
> > [1] William Peebles, & Saining Xie. (2023). Scalable Diffusion Models with Transformers.
> >
> > [2] Alexey Dosovitskiy, Lucas Beyer, Alexander Kolesnikov, Dirk Weissenborn, Xiaohua Zhai, Thomas Unterthiner, Mostafa Dehghani, Matthias Minderer, Georg Heigold, Sylvain Gelly, Jakob Uszkoreit, & Neil Houlsby. (2021). An Image is Worth 16x16 Words: Transformers for Image Recognition at Scale.
> >
> > [3] Kevin Black, et al. (2024). $π_0$: A Vision-Language-Action Flow Model for General Robot Control.
> >
> > [4] Benjamin Rhodes, Sander Vandenhaute, Vaidotas Šimkus, James Gin, Jonathan Godwin, Tim Duignan, & Mark Neumann. (2025). Orb-v3: atomistic simulation at scale.
> >
> > [5] Eric Qu, & Aditi S. Krishnapriyan. (2024). The Importance of Being Scalable: Improving the Speed and Accuracy of Neural Network Interatomic Potentials Across Chemical Domains.
> >
> > [6] Anthony Brohan, et al. (2023). RT-1: Robotics Transformer for Real-World Control at Scale.

---

### Official Review · Reviewer_vmJS · 2025-10-30

**Soundness:** 2
**Presentation:** 2
**Contribution:** 2
**Rating:** 4
**Confidence:** 5

**Summary:**

This paper investigates the necessity of graph neural networks (GNNs) and their inherent graph inductive biases for modeling molecular systems, specifically for machine learning interatomic potentials (MLIPs) that predict energies and forces. The authors challenge the dominance of GNNs by proposing the use of a standard, unmodified Transformer architecture trained directly on atomic Cartesian coordinates, without any predefined molecular graph or explicit physical priors (like equivariance). Using the OMo25 dataset, they demonstrate that a 1B parameter Transformer can achieve competitive accuracy (energy and force MAE) compared to a state-of-the-art 6M parameter equivariant GNN (eSEN) under a matched training compute budget. Analysis of the trained Transformer reveals that it learns physically meaningful patterns, such as attention weights decaying with distance and adaptive attention radii based on local atomic density, effectively learning graph-like locality without being given a graph. Furthermore, the study shows that the Transformer exhibits predictable scaling behavior consistent with observations in other domains, suggesting potential for further improvement with scale, contrasting with reported difficulties in scaling GNNs . The authors argue these findings suggest graph biases might be learned implicitly from data, pointing towards standardized, scalable architectures for molecular modeling .

**Strengths:**

- The work addresses a core question about the necessity of explicit graph structure and associated inductive biases in molecular ML, challenging the prevailing GNN paradigm. Exploring the capabilities of general-purpose architectures like Transformers in this domain is timely and important.
- Using a standard Transformer with minimal modifications provides a clean testbed to study the emergence of relational patterns directly from data, without confounding factors from complex, hand-engineered modules.
- Achieving competitive accuracy against a SOTA GNN (designed with strong physical priors) using a generic Transformer under matched compute is a significant and perhaps surprising result. It strongly suggests that sufficient data and model capacity might compensate for the lack of explicit biases.

**Weaknesses:**

- While compute-matched performance is impressive, the Transformer (**1B** params) is vastly larger than the GNN (**6M** params). This raises questions about data efficiency: the Transformer might require significantly more data or computation per parameter update to learn the necessary patterns that GNNs get "for free" from their architecture.
- The experiments are primarily conducted on the OMo125 dataset. How well does the graph-free Transformer generalize to different chemical spaces, system sizes, or tasks (e.g., predicting properties other than energy/force) compared to GNNs on the standard benchmark like QM9 and MD17? GNNs' graph bias might aid OOD generalization , although the paper also notes potential GNN generalization issues.

**Questions:**

- Could the authors comment further on the data efficiency trade-off? Is the large parameter count primarily needed to learn the relational structure that GNNs encode explicitly?
- It would be valuable to see experiments on generalization, perhaps transferring a trained model to a related but distinct dataset (e.g., QM9) or evaluating on molecules significantly larger than those seen during training.

---

> ### Author Response · Authors · 2025-11-24
>
> We respond to individual points below.
>
> > While compute-matched performance is impressive, the Transformer (1B params) is vastly larger than the GNN (6M params). This raises questions about data efficiency: the Transformer might require significantly more data or computation per parameter update to learn the necessary patterns that GNNs get "for free" from their architecture.
>
> We note that we use the **exact same training data** as the 6M parameter GNN: the OMol 4M training split. As the reviewer notes, our training is compute matched, that is, the total number of training FLOPs is controlled to be the same between both models. This means that the 1B parameter model uses **fewer** FLOPs per parameter than the 6M model, not more.
>
>
> > How well does the graph-free Transformer generalize to different chemical spaces, system sizes, or tasks (e.g., predicting properties other than energy/force) compared to GNNs on the standard benchmark like QM9 and MD17? GNNs' graph bias might aid OOD generalization , although the paper also notes potential GNN generalization issues.
>
> OMol25 is the most diverse computational chemistry dataset available to date, consisting of 100M+ molecular structures computed at a much higher accuracy than datasets like QM9 and MD17. The MLIP community has generally shifted to this being one of the main standard benchmarks, and the recent state-of-the-art MLIPs are trained and evaluated here [1]. QM9 and MD17 are older datasets calculated at lower levels of theory, and are now much less relevant to the computational chemistry community. They are also small compared to OMol25, both in terms of quantity/diversity but also in terms of system size; therefore the community has largely moved on to evaluating on these more accurate and more diverse datasets. Note that models trained on OMol25 seem to actually capture important scientific phenomena [1][5][6], much more so than models trained on prior datasets.
>
> Nevertheless, we present a clearer evaluation of system generalization by reporting the breakdown of performance (relative to the mean performance of the model) across the different splits in OMol25:
> | Split                                             | Biomolecules    |  Electrolytes | Metal Complexes | Neutral Organics |
> |---------------------------------------------------|-----------------|:-------------:|:---------------:|------------------|
> | Percent Difference From Mean (eSEN 6M/ Transformer) | -37.59 / -29.79 | -2.84 / -4.89 | 210.84 / 71.34  | 55.03 / 52.3     |
>
> **Note that the validation split in OMol25 comprises of unseen compositions and is the largest and most diverse molecular evaluation set to date.** The Transformer has consistent performance across the diverse unseen systems (compared to the reference GNN). Similar to the GNN, it performs slightly better on the larger systems (biomolecules), and slightly worse on the metal complexes, likely due to the high spin/charge systems.

---

> > ### Author Response · Authors · 2025-11-24
> >
> > > Could the authors comment further on the data efficiency trade-off? Is the large parameter count primarily needed to learn the relational structure that GNNs encode explicitly?
> >
> > At the current scales of data and compute available to the community, we showed that a Transformer can match the energy and force errors of a GNN using the same dataset and training compute budget (4M structures from OMol 4M and O(10^20) FLOPs). As noted by the reviewer, the Transformer needs more parameters to reach that level of performance, and yes this is due to the fact that Transformers have minimal inductive biases: in general a model with many inductive biases (i.e. Newton’s laws of motion, SVM with custom kernels, etc.) might require fewer parameters (i.e. just the gravitational constant G) compared to a model with few inductive biases since the parameters need to learn patterns from data directly. For example, in our case we find that the Transformer is able to learn things like rotational equivariance and locality from the data alone.
> >
> > Importantly, since the operation performed by these parameters are simple and efficiently supported on modern hardware and software, the 1B parameter Transformer is faster than the 6M parameter equivariant GNN despite having so many more parameters (see Apx B.2 for more discussion of these points).
> >
> > We also emphasize that it can be desirable to learn inductive biases from data. When the hypothesis class is sufficiently expressive and there is enough data and compute, allowing a model to discover useful structures through learning and search can lead to more general and flexible solutions than hardcoding biases by hand (see Sec. 4.3, Sec. 5, Fig. 12). This does not imply that unconstrained models are always optimal: for extremely data-scarce problems, simpler models with inductive biases (even non-deep learning approaches such as sGDML [3]) or classical force fields may still be preferable. Nevertheless, as demonstrated throughout the broader deep learning community, showing that unconstrained models can serve as scalable representation learners for MLIPs and molecular modeling enables us to exploit growing data and compute resources, while also providing a foundation for downstream fine-tuning and adaptation.
> >
> > > It would be valuable to see experiments on generalization, perhaps transferring a trained model to a related but distinct dataset (e.g., QM9) or evaluating on molecules significantly larger than those seen during training.
> >
> > We report performance on the PCQM4Mv2 benchmark [4]. The unmodified Transformer performs competitively with graph-based models and is only outperformed by architectures with design choices and training strategies specifically developed for this benchmark (indicated by a *).
> >
> >
> > | Model                  | EGT+Tri. Attn.+RDKit Coords. (*) | GraphGPT(MLM tv0) (*)|   GPS  | TokenGT (Lap) | GIN    | GCN    | Transformer (ours) |
> > |------------------------|------------------------------|:-----------------:|:------:|---------------|--------|--------|-------------|
> > | HOMO-LUMO Gap MAE (eV) | 0.068                       | 0.080            | 0.086 | 0.092        | 0.122 | 0.140 | 0.086    |
> >
> >
> >
> > [1] Daniel S. Levine, Muhammed Shuaibi, Evan Walter Clark Spotte-Smith, Michael G. Taylor, Muhammad R. Hasyim, Kyle Michel, Ilyes Batatia, Gábor Csányi, Misko Dzamba, Peter Eastman, Nathan C. Frey, Xiang Fu, Vahe Gharakhanyan, Aditi S. Krishnapriyan, Joshua A. Rackers, Sanjeev Raja, Ammar Rizvi, Andrew S. Rosen, Zachary Ulissi, Santiago Vargas, C. Lawrence Zitnick, Samuel M. Blau, & Brandon M. Wood. (2025). The Open Molecules 2025 (OMol25) Dataset, Evaluations, and Models.
> >
> > [2] Richard Sutton, 2019. URL http://www.incompleteideas.net/IncIdeas/
> > BitterLesson.html.
> >
> > [3] Chmiela, S., Sauceda, H., Poltavsky, I., Müller, K.R., & Tkatchenko, A. (2019). sGDML: Constructing accurate and data efficient molecular force fields using machine learning. Computer Physics Communications, 240, 38–45.
> >
> > [4] Weihua Hu, Matthias Fey, Marinka Zitnik, Yuxiao Dong, Hongyu Ren, Bowen Liu, Michele Catasta, & Jure Leskovec. (2021). Open Graph Benchmark: Datasets for Machine Learning on Graphs.
> >
> > [5] Brandon M. Wood, Misko Dzamba, Xiang Fu, Meng Gao, Muhammed Shuaibi, Luis Barroso-Luque, Kareem Abdelmaqsoud, Vahe Gharakhanyan, John R. Kitchin, Daniel S. Levine, Kyle Michel, Anuroop Sriram, Taco Cohen, Abhishek Das, Ammar Rizvi, Sushree Jagriti Sahoo, Zachary W. Ulissi, & C. Lawrence Zitnick. (2025). UMA: A Family of Universal Models for Atoms.
> >
> > [6] VanZanten S, Wagen C. Benchmarking OMol25-Trained Models on Experimental Reduction-Potential and Electron-Affinity Data. ChemRxiv. 2025; doi:10.26434/chemrxiv-2025-3stpx This content is a preprint and has not been peer-reviewed.

---

### Official Review · Reviewer_mKVr · 2025-10-31

**Soundness:** 3
**Presentation:** 3
**Contribution:** 2
**Rating:** 2
**Confidence:** 4

**Summary:**

Graph Neural Networks (GNNs) have shown promise for modeling complex molecular dynamics and relationships. These remain the default architecture for drug discovery tasks. However, GNNs operate on pre-defined graphs which are hard-coded to make learning amenable. The paper empirically studies and compares the behavior of pure transformers, without graph priors, with GNNs for the task of Machine Learning Interatomic Potentials (MLIP). Authors show that pure transformers, when trained on discretized and binned tokens representing xyz cartesian coordinates, energy and forces, perform similarly as state of the art equivariant GNNs. Transformeres are found to be faster and more efficient and obey scaling laws of compute and budget. Furthermore, the paper takes an interpretability approach trying to understand representations learned by the model. Transformer captures local features in its early layers while later layers capture global layers. Additionally, the paper uncovers variation of attention with radius showing that the model adapts its attention pattern as per the radius per atom and packing of atoms in its neighborhood.

**Strengths:**

* The paper is well written and organized.
* The paper studies and disects the model from an intuitive perspective.

**Weaknesses:**

* **Contribution:** My main concern is the central contribution and direction of the paper. It is unclear as to what new knowledge and insights the paper is adding. The paper assesses and compares the vanilla transformer architecture with equivariant GNNs for MLIP. However, the paper limits itself to results, trends and insights that have already been established. It is well known that transformers scale and obey scaling laws, irrespective of the type and complexity of data. These results were obtained both theoreticaly and empirically [1, 2] and corresponding to applications to drug design [3]. The paper studies scaling laws and performance of transformers which has been well demosntrated in the past. Furthermore, inference speeds and latency studies of the transformer block have also drawn insights into its optimized operation [4]. At its core, the paper limits itself to a naive application of the transformer architecture to MLIP.

* **Comparison to GNN:** The paper compares a 1B transformer with a state of the art equivariant GNN. However, this comparison remains unclear. It is well known that GNNs also scale and obey scaling laws [5, 6], both in compute and budget. A more thorough evaluation would consider scaling both the transformer and GNN and comparing their scaling trends. Furthermore, while transformer a larger transformers is faster at inference and memory, GNNs are also found to be more efficient when distributing computation. Hence, a fair comparison would consider utilizing the same hardware and training setting for equal parameters and FLOP constraints for different datasets. An additional aspect to note is that GNNs are performant out of the box for both supervised and unsupervised data and handle sparse inputs effectively. An analogous comparison with the transformer model would help shed light on its limitations. Finally, GNNs are more compact (1-100M) even for state of the art models and do not require additional tricks such as KV caching and speculative decoding at inference. Thus, a comprehensive comparison between transformer and GNNs is found to be missing for MLIP.

* **Experiments:** The paper assesses how transformers scale and behave. However, the paper does not answer why they behave in the observed ways. We know that GNNs often fail due to oversquashing. But the paper does not study how transformers hallucinate or beave differently for molecular inputs. The paper aims to uncover their representational aspects but does not take an interpretability-based approach towards the architecture. Instead, authors crudely explain transformer outputs and layerwise representations. It would be worthwhile to evaluate any discovered circuits within the model or a grouping of neurons that specifically stores certain features. Furthermore, authors could help explain activation patterns and the distribution of weights within the residual stream for different molcular types.

[1]. Kaplan et al, Scaling Laws for Neural Language Models, https://arxiv.org/abs/2001.08361.
[2]. Zhang et al, When Scaling Meets LLM Finetuning: The Effect of Data, Model and Finetuning Method, https://arxiv.org/html/2402.17193v1.
[3]. Abramson et al, Accurate structure prediction of biomolecular interactions with AlphaFold 3, Nature, 2024.
[4]. Austin et al, How To Scale Your Model, https://jax-ml.github.io/scaling-book/.
[5]. Sypetkowski et al, On the Scalability of GNNs for Molecular Graphs, NeurIPS 2024.
[6]. Shirzad et al, Exphormer: Scaling Graph Transformers with Expander Graphs, ICML 2023.

**Questions:**

Refer to weaknesses

---

> ### Author Response · Authors · 2025-11-24
>
> We are confused due to a number of comments being unrelated to the contents of our paper. We reply to individual points below:
>
> > It is unclear as to what new knowledge and insights the paper is adding. The paper limits itself to results, trends and insights that have already been established.
>
> The field of molecular modeling with ML, and especially the field of machine learning interatomic potentials (MLIPs), has largely relied on using GNNs. To our knowledge, there is no prior work that uses an unmodified Transformer as an MLIP. While broader molecular property prediction work has explored varied architectures, some of which take inspiration from the Transformer, many of these works make substantial modifications to the standard Transformer, for example by modifying the attention mechanism or adding graph-based featurization (see our related works for more discussion). We believe that examining to what extent an unmodified Transformer is capable of learning molecular representations is therefore an interesting way to build a bridge between the field of molecular ML (which has relied on more domain specific architectures) to a large part of the deep learning community that has largely converged to the Transformer.
>
>
> > It is well known that transformers scale and obey scaling laws, irrespective of the type and complexity of data. These results were obtained both theoretically and empirically [1, 2] and corresponding to applications to drug design [3].
>
> We are not claiming to be the first to establish scaling laws for Transformers. As far as we know, however, we are the first to establish scaling laws for Transformers in the field of MLIPs. Although the reviewer notes the AlphaFold3 paper, the area of ML for structural biology is a completely application area and field from MLIPs. AlphaFold3 also uses a custom architecture (not an unmodified Transformer) and does not perform a scaling law analysis as far as we can tell.
>
> > At its core, the paper limits itself to a naive application of the transformer architecture to MLIP.
>
> The MLIP field has been moving very quickly with a lot of new work over the past few years. Since the field has largely relied on the GNN architecture, we believe that this is an interesting investigation to see if an unmodified Transformer, without a graph prior, can approximate reference energies and forces. Previous work in this spirit has gained a lot of attention in the field, for example investigating the role of rotational equivariance [1][2], and our work takes it one step further by investigating the role of the graph prior. Our investigation reveals that many of the favorable properties in GNNs can emerge adaptively in Transformers, while still offering competitive performance on the OMol25 dataset, both of which are interesting for the molecular ML community.
>
> > It is well known that GNNs also scale and obey scaling laws [5, 6], both in compute and budget.
>
> As noted in our related works, we are aware that people have conducted scaling analyses with graph neural networks and MLIPs. However, these previous scaling studies, including the reference noted by the reviewer (see Fig.2 and Sec 4.1 in [5] from reviewer) have observed significant deviations from consistent powerlaw relationships. We discuss in more detail some of the work from the broader community, both theoretical and practical, in our related works section. We do not claim that GNNs cannot scale. We acknowledge the substantial progress in the community while summarizing why consistent GNN scaling has been difficult to establish in prior work. Our study focuses on ML for 3D molecular geometries, where, as far as we are aware, there are no widely used GNNs at the scale of the current largest Transformers (hundreds of billions of parameters).
>
> > A more thorough evaluation would consider scaling both the transformer and GNN and comparing their scaling trends.
>
> We will add the following result where we perform a data scaling analysis with the UMA model, which is one of the largest GNN-based MLIPs [5]. We train models for varying numbers of steps and calculate a data scaling exponent. We hope this direct comparison provides further clarification on our results.
>
>
> | Model                 | UMA S | Transformer 5M | Transformer 28M | Transformer 90M | Transformer 306M |
> |-----------------------|-------|:--------------:|:---------------:|-----------------|------------------|
> | Data Scaling Exponent | -0.38 | -0.24          | -0.35           | -0.51           | -0.51            |

---

> > ### Author Response · Authors · 2025-11-24
> >
> > > Furthermore, while transformer a larger transformers is faster at inference and memory, GNNs are also found to be more efficient when distributing computation. Hence, a fair comparison would consider utilizing the same hardware and training setting for equal parameters and FLOP constraints for different datasets.
> >
> > Could the reviewer clarify what specific comparison would make the paper stronger? We already do a compute-matched comparison on the OMol25 datasets, which is widely regarded as the most relevant molecular dataset for the MLIP community, which is the focus of our work. We also already compare the training and inference speeds on this dataset using the same hardware for both models.
> >
> > > An additional aspect to note is that GNNs are performant out of the box for both supervised and unsupervised data and handle sparse inputs effectively. An analogous comparison with the transformer model would help shed light on its limitations.
> >
> > Could the reviewer clarify what specific comparison they believe would make the paper stronger?
> >
> > > Finally, GNNs are more compact (1-100M) even for state of the art models and do not require additional tricks such as KV caching and speculative decoding at inference.
> >
> > KV caching and speculative decoding are methods used for autoregressive generation and ***do not apply in the context of our work.*** Our final model is a bi-directional Transformer.
> >
> > > Thus, a comprehensive comparison between transformer and GNNs is found to be missing for MLIP.
> >
> > We appreciate the effort from the reviewer and are open to collaborating with the reviewer to make the paper stronger. However, we would appreciate more concreteness with regards to what exact comparison would make the paper stronger. We make a compute-matched comparison to a state-of-the-art GNN on the recently released and widely accepted OMol25 dataset, which is the most comprehensive molecular dataset to date. Please let us know what exact axis you think would make the comparison clearer.
> >
> > > We know that GNNs often fail due to oversquashing. But the paper does not study how transformers hallucinate or beave differently for molecular inputs.
> >
> > Could the reviewer clarify what this means? We emphasize that we are not using a Transformer for autoregressive generation in our work, so we are not sure what the reviewer means by hallucination.
> >
> > > The paper aims to uncover their representational aspects but does not take an interpretability-based approach towards the architecture. Instead, authors crudely explain transformer outputs and layerwise representations.
> >
> > Could the reviewer clarify concretely what they believe would make the paper stronger? We note that studying attention maps is a standard practice among, for example, previous vision transformer works [3,4]. A full interpretability study is likely a whole project in and of itself, with whole subfields of ML dedicated to this study.
> >
> > [1] Benjamin Rhodes, Sander Vandenhaute, Vaidotas Šimkus, James Gin, Jonathan Godwin, Tim Duignan, & Mark Neumann. (2025). Orb-v3: atomistic simulation at scale.
> >
> > [2] Eric Qu, & Aditi S. Krishnapriyan. (2024). The Importance of Being Scalable: Improving the Speed and Accuracy of Neural Network Interatomic Potentials Across Chemical Domains.
> >
> > [3] Alexey Dosovitskiy, Lucas Beyer, Alexander Kolesnikov, Dirk Weissenborn, Xiaohua Zhai, Thomas Unterthiner, Mostafa Dehghani, Matthias Minderer, Georg Heigold, Sylvain Gelly, Jakob Uszkoreit, & Neil Houlsby. (2021). An Image is Worth 16x16 Words: Transformers for Image Recognition at Scale.
> >
> > [4] Timothée Darcet, Maxime Oquab, Julien Mairal, & Piotr Bojanowski. (2024). Vision Transformers Need Registers.
> >
> > [5] Brandon M. Wood, Misko Dzamba, Xiang Fu, Meng Gao, Muhammed Shuaibi, Luis Barroso-Luque, Kareem Abdelmaqsoud, Vahe Gharakhanyan, John R. Kitchin, Daniel S. Levine, Kyle Michel, Anuroop Sriram, Taco Cohen, Abhishek Das, Ammar Rizvi, Sushree Jagriti Sahoo, Zachary W. Ulissi, & C. Lawrence Zitnick. (2025). UMA: A Family of Universal Models for Atoms.

---

### Official Review · Reviewer_U69B · 2025-11-01

**Soundness:** 4
**Presentation:** 3
**Contribution:** 3
**Rating:** 8
**Confidence:** 4

**Summary:**

This paper investigates the use of Transformer models for molecular property prediction, specifically energy and force estimation, without relying on the graph-based inductive biases traditionally used in Graph Neural Networks (GNNs). The authors train a Transformer directly on Cartesian coordinates and analyze its performance on the OMol25 dataset. This research challenges the prevailing approach of using GNNs with predefined graph structures and presents a compelling argument that Transformers, when appropriately scaled, can achieve competitive performance for molecular modeling tasks.

**Strengths:**

### Novel Approach:
The most significant contribution of this work is the application of unmodified Transformers for molecular modeling. By removing graph-based priors, the authors present a more flexible architecture that can potentially scale better and handle a wider range of molecular systems. This is a fresh direction that could influence future molecular property prediction models.

### Competitive Results:
Despite lacking explicit geometric and physical priors, the Transformer model achieves energy and force errors comparable to state-of-the-art equivariant GNNs on the OMol25 dataset, under the same computational budget. This demonstrates that the Transformer can effectively learn the relational and physical patterns in molecular data.

### Scalability and Efficiency:
The paper highlights the scalability of Transformers, noting their ability to perform well with large datasets and scale predictably with training resources. This scaling behavior, backed by empirical results, mirrors findings in other domains of machine learning, such as natural language processing, and suggests that similar approaches can be adopted in the molecular modeling community.

### Insightful Representation Analysis:
The authors perform a thorough investigation of the learned representations, especially the attention maps, and observe that the Transformer naturally learns relationships akin to those found in traditional GNNs. The distance-dependent attention and adaptability to atomic environments are key findings that suggest Transformers can capture complex, spatially localized interactions without needing predefined graphs.

### Potential for Broader Applications:
The paper emphasizes that the Transformer framework could generalize to other chemical systems and could eventually integrate with multi-modal data sources, further enhancing its versatility. The authors also suggest avenues for future research, such as integrating physical constraints and improving the generalization of the model across diverse molecular environments.

**Weaknesses:**

### Lack of Strict Physical Constraints:
While the Transformer model is successful in approximating energies and forces, it may not adhere to all strict physical principles, which is a limitation for some scientific applications where precise physical fidelity is required. The authors acknowledge this and propose future work to integrate constraints or fine-tune the model, but the current approach may not be suitable for all use cases without such modifications.
### Limited Evaluation in Larger Molecule Systems:
While the model is tested on the OMol25 dataset and achieves competitive results, this benchmark may not fully capture the complexities of larger molecular systems. A more extensive evaluation, particularly in dynamic simulations and larger-scale systems, would offer a clearer understanding of the model's general applicability and robustness across different types of molecular environments.

**Questions:**

1. Model Performance and Comparison

How does the performance of the Transformer model compare to that of GNNs when tested on other molecular datasets beyond OMol25?

What are the implications of the Transformer model’s speed advantages in terms of computational cost and practicality for large-scale molecular simulations?

2. Generalization and Scalability

While the Transformer achieves competitive performance with GNNs, how does its generalization ability perform when applied to new molecular geometries or diverse chemical systems that differ from those in the OMol25 dataset?

Given the scaling behavior observed with the Transformer, what might be the challenges of scaling this model beyond 1B parameters, and are there limitations to this scalability?

3. Physical Constraints

The paper discusses potential limitations in adhering to strict physical laws. What are the specific physical principles that the Transformer might struggle to respect, and how could these be addressed in future research?

Would a hybrid model combining the flexibility of Transformers with physics-based constraints offer a more accurate solution for molecular dynamics simulations?

**Details Of Ethics Concerns:**

No Ethics Concerns

---

> ### Author Response · Authors · 2025-11-24
>
> We thank the reviewer for the detailed feedback. We appreciate the careful evaluation of our work and the interesting questions. We reply to individual points below:
>
> > Physical constraints
>
> Although the focus of this work was investigating the role of the graph-prior, we agree that it is important to rigorously evaluate models before applying them broadly to downstream scientific discovery. In line with previous work [1][2], we observed that the Transformer was able to almost perfectly learn rotational equivariance (Sec. 4.1), and the model can also be fine-tuned to predict a conservative force (Fig. 7). There is ample evidence in the MLIP field and the broader deep learning community ([3][4][5] to name a few) suggesting that training strategies can help models learn constraints from data, and we think it is an interesting direction for future work to further explore this with Transformers.
>
> We also emphasize that it can be desirable to learn inductive biases from data. When the hypothesis class is sufficiently expressive and there is enough data and compute, allowing a model to discover useful structures through learning and search can lead to more general and flexible solutions than hardcoding biases by hand (see Sec. 4.3, Sec. 5, Fig. 12). This does not imply that unconstrained models are always optimal: for extremely data-scarce problems, simpler models with inductive biases (even non-deep learning approaches such as sGDML [9]) or classical force fields may still be preferable. Nevertheless, as demonstrated throughout the broader deep learning community, showing that unconstrained models can serve as scalable representation learners for MLIPs and molecular modeling enables us to exploit growing data and compute resources, while also providing a foundation for downstream fine-tuning and adaptation.
>
>
> > What are the implications of the Transformer model’s speed advantages in terms of computational cost and practicality for large-scale molecular simulations?
>
> This is an interesting question. Although the training was compute matched, the 1B Transformer does train and run inference faster than the 6M GNN. We think it is an interesting direction for future work to push the length and time scales of MD simulations by leveraging the mature hardware and software ecosystem for Transformers.
>
> > Given the scaling behavior observed with the Transformer, what might be the challenges of scaling this model beyond 1B parameters, and are there limitations to this scalability?
>
> It is certainly resource intensive to train models at scale, and while we can’t guarantee that the scaling behavior will continue to hold, evidence from other fields of deep learning [6][7] paired with our scaling laws provide evidence that consistent improvements could continue to be observed: the largest current deep learning models are trained with multiple orders of magnitude more parameters, data, and compute that what was trained in this paper.
>
> > How does the performance of the Transformer model compare to that of GNNs when tested on other molecular datasets beyond OMol25?
>
> While we focused our analysis on the OMol dataset, since this is the largest and most diverse computational chemistry dataset to date, we now additionally report performance on the PCQM4Mv2 benchmark [8]. The unmodified Transformer performs competitively with graph-based models and is only outperformed by architectures with design choices and training strategies specifically developed for this benchmark (indicated by a *).
>
>
> | Model                  | EGT+Tri. Attn.+RDKit Coords. (*) | GraphGPT(MLM tv0) (*)|   GPS  | TokenGT (Lap) | GIN    | GCN    | Transformer (ours) |
> |------------------------|------------------------------|:-----------------:|:------:|---------------|--------|--------|-------------|
> | HOMO-LUMO Gap MAE (eV) | 0.068                       | 0.080            | 0.086 | 0.092        | 0.122 | 0.140 | 0.086    |

---

> > ### Author Response · Authors · 2025-11-24
> >
> > > While the Transformer achieves competitive performance with GNNs, how does its generalization ability perform when applied to new molecular geometries or diverse chemical systems that differ from those in the OMol25 dataset?
> >
> > Thank you for the question. We note that the validation split in OMol comprises unseen compositions and is the largest and most diverse molecular evaluation set to date (100M+ samples). We present a clearer evaluation of system generalization by reporting the breakdown of performance (relative to the mean performance of the model) across the different splits in OMol:
> > | Split                                             | Biomolecules    |  Electrolytes | Metal Complexes | Neutral Organics |
> > |---------------------------------------------------|-----------------|:-------------:|:---------------:|------------------|
> > | Percent Difference From Mean (eSEN 6M/ Transformer) | -37.59 / -29.79 | -2.84 / -4.89 | 210.84 / 71.34  | 55.03 / 52.3     |
> >
> > The Transformer has consistent performance across the diverse unseen systems (compared to the reference GNN). Similar to the GNN, it performs slightly better on the larger systems (biomolecules), and slightly worse on the metal complexes, likely due to the high spin/charge systems.
> >
> > While we focused this work on investigating to what extent models without physical inductive biases are able to learn molecular representations, we agree with the reviewer that further rigorous downstream evaluations are important for the community. We plan to do these in future work before using these types of models broadly in downstream scientific applications.
> >
> >
> > [1] Benjamin Rhodes, Sander Vandenhaute, Vaidotas Šimkus, James Gin, Jonathan Godwin, Tim Duignan, & Mark Neumann. (2025). Orb-v3: atomistic simulation at scale.
> >
> > [2] Eric Qu, & Aditi S. Krishnapriyan. (2024). The Importance of Being Scalable: Improving the Speed and Accuracy of Neural Network Interatomic Potentials Across Chemical Domains.
> >
> > [3] Bowen Deng, Yunyeong Choi, Peichen Zhong, Janosh Riebesell, Shashwat Anand, Zhuohan Li, KyuJung Jun, Kristin A. Persson, & Gerbrand Ceder. (2024). Overcoming systematic softening in universal machine learning interatomic potentials by fine-tuning.
> >
> > [4] Rafael Rafailov, Archit Sharma, Eric Mitchell, Stefano Ermon, Christopher D. Manning, & Chelsea Finn. (2024). Direct Preference Optimization: Your Language Model is Secretly a Reward Model.
> >
> > [5] Thaddäus Wiedemer, Yuxuan Li, Paul Vicol, Shixiang Shane Gu, Nick Matarese, Kevin Swersky, Been Kim, Priyank Jaini, & Robert Geirhos. (2025). Video models are zero-shot learners and reasoners.
> >
> > [6] Tom B. Brown, et al. (2020). Language Models are Few-Shot Learners.
> >
> > [7] Jared Kaplan, Sam McCandlish, Tom Henighan, Tom B. Brown, Benjamin Chess, Rewon Child, Scott Gray, Alec Radford, Jeffrey Wu, & Dario Amodei. (2020). Scaling Laws for Neural Language Models.
> >
> > [8] Weihua Hu, Matthias Fey, Marinka Zitnik, Yuxiao Dong, Hongyu Ren, Bowen Liu, Michele Catasta, & Jure Leskovec. (2021). Open Graph Benchmark: Datasets for Machine Learning on Graphs.
> >
> > [9] Chmiela, S., Sauceda, H., Poltavsky, I., Müller, K.R., & Tkatchenko, A. (2019). sGDML: Constructing accurate and data efficient molecular force fields using machine learning. Computer Physics Communications, 240, 38–45.

---

### Author Response · Authors · 2025-11-24

We are open to working with reviewers to make our paper stronger based on constructive feedback. However, there are a number of misunderstandings and incorrect claims that do not seem to actually be related to our paper. We directly quote some of these comments verbatim here, and we also reply in detail to individual reviewers below.

> “We know that GNNs often fail due to oversquashing. But the paper does not study how transformers hallucinate”
> “[GNNs] do not require additional tricks such as KV caching and speculative decoding”

We emphasize that there is no generation in our experiments so things like “hallucination”, “KV caching”, and “speculative decoding” are not relevant for our work. In general, there are a number of citations and comments from reviewers that reference LLM literature (chain-of-thought reasoning, LLM fine-tuning, etc.) which are not at all related to our use of a bi-direcitonal Transformer architecture to perform energy and force regression.

> “effectiveness [of the Transformer] in continuous space is still in doubt”

We emphasize that Transformers have been successfully used for regression/continuous tasks across diffusion, vision, robotics, and beyond [1][2][3].

> “Results of this paper are likely due to large-scale data training, which has little significance in the field of science”

This is one of the main themes of deep learning: many of the successes of deep learning in general are based on figuring out how to effectively do large-scale training. This is also the central investigation of our paper: in the last year, the ML interatomic potentials / ML for chemistry community has generated large, comprehensive molecular datasets, and we investigate how this enables training competitive models with fewer inductive biases.

> “[inductive biases reduce] training costs”
> “The Transformer might require significantly more data or computation per parameter update”, “[inductive biases reduce] training costs”
>“a fair comparison would consider utilizing the same hardware and training setting for equal parameters and FLOP constraints”

In the main experiment in our paper, we explicitly show that with the same data and training computational budget, a 1B parameter Transformer can achieve competitive performance with a state-of-the-art 6M parameter GNN on OMol25, the largest and most comprehensive molecular dataset to date. This means that the Transformer requires __fewer__ data points or computation steps per parameter.

> “[Compare with] standard benchmark like QM9 and MD17”

Open Molecules 2025 (OMol25) is the most diverse and largest computational chemistry dataset available to date, with 100M+ samples. QM9 and MD17 are older, much smaller datasets calculated at lower accuracy levels, that are now much less relevant to the computational chemistry community. ***In fact, these datasets are already represented and recalculated more accurately in OMol25. OMol25 is generally considered by the MLIP community as the most comprehensive molecular benchmark.***

> “The Transformer tends to memorize the patterns of the training set rather than performing inference in many tasks [1], which is fatal for the mathematics in the molecular space.
[1] Is Chain-of-Thought Reasoning of LLMs a Mirage? A Data Distribution Lens”

The reviewer references a paper from the LLM community that is unrelated to our use of the encoder Transformer architecture to perform regression.

> “[Scaling results were already] obtained both theoretically and empirically [citation to LLM scaling works] and corresponding to applications to drug design [citation to AlphaFold3]”

Although the reviewer notes the AlphaFold3 paper, the area of ML for structural biology is a completely different application area and field from ML interatomic potentials (MLIPs) and ML for computational chemistry. AlphaFold3 also uses a custom architecture (not an unmodified Transformer) and does not perform a scaling law analysis as far as we can tell.

We are not claiming to be the first to establish scaling laws for Transformers, and discuss the LLM scaling literature in our related works. As far as we know, however, we are the first to establish scaling laws for Transformers in the field of MLIPs. We emphasize that textual and image data, where scaling has been observed previously, is very different from chemical data. Establishing scaling laws can also require specific and consistent training recipes for different data modalities, which we investigate for chemical data in this work.

As far as we know, unmodified Transformers have not been applied to the rapidly growing field of MLIPs, which is dominated by equivariant GNNs. Our investigation reveals that many of the favorable properties in GNNs can emerge adaptively in Transformers, while still offering competitive performance on OMol25.

We note again that a number of comments are either incorrect or unrelated to our paper. We remain open to discussion surrounding constructive criticism.

---

> ### Author Response · Authors · 2025-11-24
>
> [1] William Peebles, & Saining Xie. (2023). Scalable Diffusion Models with Transformers.
>
> [2] Alexey Dosovitskiy, et al. (2021). An Image is Worth 16x16 Words: Transformers for Image Recognition at Scale.
>
> [3] Kevin Black, et al. (2024). $π_0$: A Vision-Language-Action Flow Model for General Robot Control.

---

### Meta-Review · Area_Chair_9HF9 · 2026-01-07

**Summary:**

The submission reports an investigation on the behavior of a vanilla Transformer model for energy/force prediction where everything (e.g., atom coordinates, the force array) are sequentialized and without any molecular or 3D-neighborhood graph. Scaling law is observed, and the model demonstrates certain mechanism that qualitatively recovers some physical intuition.

Reviewers acknowledge the innovation to consider studying this unusual choice of modeling, the relevance of answering a basic question in the field, and some seemingly positive performance outcome.

Reviewers' concerns include:
1. Lack of physical guarantees (e.g., conservativeness, translational and rotational invariance, and permutation invariance).
2. Scaling to large molecules are not included, which would also be of interest of the community and complete the topic.
3. The targeted contribution of the study is not sufficiently clear, considering that Transformers and some GNNs are already demonstrated to follow scaling law. This should also be reflected in experiment design.

**Reviewer Concerns:**

1. Regarding physical guarantees, although the authors showed demonstrations that the model could learn such behaviors from sufficient data (but not for permutation invariance), it is still not an ultimately meaningful demonstration for real use cases, where the model may encounter out-of-distribution samples and any deviation from the physical constraints may eventually lead to qualitatively incorrect conclusions. To make this study meaningful, I would suggest to address the users' deepest concern by e.g., running an MD simulation. If MD results leads to correct observations, then the next question would be what are the advantages of such an approach considering that it requires quite dense training data to work.

2. The authors provided experiments on the PCQM4Mv2 dataset. I appreciate the effort, and acknowledge the relevance given that there are larger molecules in this experiment, but I would further expect a more detailed study e.g. the trend and order of performance with the increase of molecule sizes. This could also present results on certain larger scales and not averaged.

3. The positioning of the contribution is made clearer in the rebuttal. Nevertheless, the meaning of the described investigation target seems a bit subjective, i.e., it may be arguable if current equivariant Transformer MLIPs are "modifications" to the vanilla Transformer in the sense that they may be the natural Transformer for processing molecular structures which lives on a manifold due to the geometric equivariance, and many of them neither involve a "graph prior". At least, I would expect the authors to address the ultimate concern that whether the learned physical constraints could support real simulation queries, which is the pre-requisite to make this study meaningful.

**Reviewer Scores:**

I would expect Reviewers mKVr and RHEQ to raise their scores, but not probably to a positive score, due to the above mentioned limitations.

---

### Decision · Program_Chairs · 2026-01-26

Reject